# How does Your RL Agent Explore?
# An Optimal Transport Analysis of Occupancy
# Measure Trajectories

## Abstract

The rising successes of RL are propelled by combining smart algorithmic strategies and deep architectures to optimize the distribution of returns and visitations over the state-action space. A quantitative framework to compare the learning processes of these eclectic RL algorithms is currently absent but desired in practice. We address this gap by representing the learning process of an RL algorithm as a sequence of policies generated during training, and then studying the policy trajectory induced in the manifold of state-action occupancy measures. Using an optimal transport-based metric, we measure the length of the paths induced by the policy sequence yielded by an RL algorithm between an initial policy and a final optimal policy. Hence, we first define the *Effort of Sequential Learning (ESL)*. ESL quantifies the relative distance that an RL algorithm travels compared to the shortest path from the initial to the optimal policy. Further, we connect the dynamics of policies in the occupancy measure space and regret (another metric to understand the suboptimality of an RL algorithm), by defining the *Optimal Movement Ratio (OMR)*. OMR assesses the fraction of movements in the occupancy measure space that effectively reduce an analogue of regret. Finally, we derive approximation guarantees to estimate ESL and OMR with finite number of samples and without access to an optimal policy. Through empirical analyses across various environments and algorithms, we demonstrate that ESL and OMR provide insights into the exploration processes of RL algorithms and hardness of different tasks in discrete and continuous MDPs.

## 1 Introduction

In recent years, significant advancements in Reinforcement Learning (RL) have been achieved in developing exploration techniques that improve learning (Bellemare et al., 2016; Burda et al., 2019; Eysenbach et al., 2019) along with new learning methods (Lazaridis et al., 2020; Müller et al., 2021; Li, 2023). With growing computational resources, these techniques have led to various successful applications of RL, such as playing games up to human proficiency (Silver et al., 2017; Jaderberg et al., 2019), controlling robots (Ibarz et al., 2021; Kaufmann et al., 2023), tuning databases and computer systems (Wang et al., 2021; Basu et al., 2019), etc. However, there remains a lack of consensus over approaches that can quantitatively compare these exploratory processes across RL algorithms and tasks (Seijen et al., 2020; Amin et al., 2021; Ladosz et al., 2022). This is attributed to some methods being algorithm-specific (Tang et al., 2017), while others provide theoretical guarantees for very specific settings (Lattimore & Szepesvári, 2020; Agarwal et al., 2022). Thus, comparing the exploratory processes of these eclectic algorithms across the multi-directional space of RL algorithm design, emerges as a natural question. However, the present literature lacks a metric to compare them except regret, which is often hard to estimate (Ramos et al., 2017; 2018).

This paper aims to address this gap based on two key observations. *First*, we observe from the linear programming formulation of RL that solving the value maximization problem is equivalent to finding an optimal occupancy measure (Syed et al., 2008; Neu & Pike-Burke, 2020; Kalagarla et al., 2021). Occupancy measure is the distribution of state-action pair visits induced by a policy (Altman, 1999; Laroche & des Combes, 2023). Under mild assumptions, a policy maps uniquely to an occupancy measure. *Second*, we observe that any RL algorithm learns by sequentially updating

policies starting from an initial policy to reach an optimal policy. The search for an optimal policy is influenced by the exploration-exploitation strategy and functional approximators, both of which impact the overall performance of the agent by determining the quality of experiences from which it learns (Zhang et al., 2019; Ladosz et al., 2022). Hereby, we term collectively the learning strategy and the exploration-exploitation interplay as the *exploratory process*.

**Contributions.** *1. A Framework.* Motivated by our observations, we abstract any RL algorithm as a trajectory of occupancy measures induced by a sequence of policies between an initial and a final (optimal) policy. The occupancy measure of a policy given an environment corresponds to the data-generating distribution of state-actions. Thus, we can quantify the effort of each policy update, i.e. the effort to shift the state-action data distributions, as the transportation distance between their occupancy measures. The total effort of learning by the algorithm can be measured as the total distance covered by its occupancy measure trajectory. We provide a mathematical basis for this quantification by proving that the space of occupancy measures is a differentiable manifold for smoothly parameterized policies (Section 3). Hence, we can compute the length of the occupancy measure trajectory on this manifold using Wasserestein distance as the metric (Villani, 2009).

*2. Effort of Sequential Learning.* In contrast to RL, if we knew the optimal policy we could update our initial policy directly via supervised or imitation learning. Effort of this learning is represented by a direct, shortest (geodesic) path from initial to optimal policy on the occupancy measure manifold. To quantify the cost of the exploratory process to learn the environment, we define the *Effort of Sequential Learning* (ESL) as the ratio of the (indirect) path traversed by an RL algorithm in the occupancy measure space to the direct distance between the initial and optimal policy (Section 3.1). Lower ESL implies more efficient exploratory process.

*3. Efforts to learn that lead to Regret-analogue minimization.* Regret is a widely used optimality measures for reward-maximizing RL algorithms (Sutton & Barto, 2018). It measures the total deviation in the value functions achieved by a sequence of policies learned by an RL algorithm with respect to the optimal algorithm that always uses the optimal policy (Sinclair et al., 2023). We show that regret is related to the sum of distances between the optimal policy and each policy in the sequence learned by the RL algorithm, in the occupancy measure space. We can define an analogue of instantaneous regret (at any one step during learning rather than cumulative), in the occupancy measure space, as the geodesic distance between the occupancy measure of the policy at this step in the learning sequence, and the optimal one. We find that not all policy updates lead to a reduction in this analogue of immediate regret, and thus define another index *Optimal Movement Ratio* that measures the fraction that do (Section 3.2).

*4. Computational and Numerical Insights.* We prove sample complexity guarantees to approximate ESL and OMR in practice as we do not have access to the occupancy measures but collection of rollouts from the corresponding policies (Section 4). We show the relation of empirical OMR and ESL to the true ones if the optimal policy is never reached by an algorithm. We conduct experiments on multiple environments, both discrete and continuous, with sparse and dense rewards, comparing state-of-the-art algorithms. We observe that by visualizing aspects of the path traversed (and by comparing ESL and OMR), we are able to compare and provide insights into their exploratory processes and the impact of task hardness on them (Section 5). The results confirm the ubiquity and effectiveness of our approach to study the exploratory processes of RL algorithms.

## 2 PRELIMINARIES

**Markov Decision Processes.** Consider an agent interacting with an environment in discrete timesteps. At each timestep $t \in \mathbb{N}$, the agent observes a state $s_t$, executes an action $a_t$, and receives a scalar reward $\mathcal{R}(s_t, a_t)$. The behaviour of the agent is defined by a policy $\pi(a_t|s_t)$, which maps the observed states to actions. The environment is modelled as a Markov Decision Process (MDP) $\mathbb{M}$ with a state space $\mathcal{S}$, action space $\mathcal{A}$, transition dynamics $\mathcal{T} : \mathcal{S} \times \mathcal{A} \to \mathcal{S}$, and reward function $\mathcal{R} : \mathcal{S} \times \mathcal{A} \to \mathbb{R}$. During task execution, the agent issues actions in response to states visited, and hence a sequence of states and actions $h_t = (s_0, a_0, s_1, a_1, ..., s_{t-1}, a_{-1}, s_t)$, here called a rollout, is observed.

In infinite-horizon settings, the state value function for a given policy $\pi$ is the expected discounted cumulative reward over time $V_\pi(s) \triangleq \mathbb{E}_\pi \left[ \sum_{t=0}^\infty \gamma^t \mathcal{R}(s_t, a_t) \mid s_0 = s \right]$, where $\gamma \in [0, 1)$ is the dis-

count rate. The goal is to learn a policy that maximises the objective $J_\mu^\pi \triangleq \mathbb{E}_{s \sim \mu}[V_\pi(s)]$, where $\mu(s)$ is the initial state distribution.

**Occupancy Measure.** The state-action occupancy measure is a distribution over the $\mathcal{S} \times \mathcal{A}$ space that represents the discounted frequency of visits to each state-action pair when executing a policy $\pi$ in the environment (Syed et al., 2008). Formally, the occupancy measure of $\pi$ is $v_\pi(s, a) \triangleq \rho \sum_{t=0}^\infty \gamma^t \mathbb{P}(s_t = s, a_t = a \mid \pi, \mu)$, where $\rho = 1 - \gamma$ is the normalizing factor.

Stationary Markovian policies allow a bijective correspondence with their state-action occupancy measures (Givchi, 2021). We express the objective $J_\mu^\pi$ in terms of the occupancy measure as

$$J_\mu^\pi = \frac{1}{\rho} \mathbb{E}_{(s,a) \sim v_\pi} \left[ \bar{\mathcal{R}}(s, a) \right], \tag{1}$$

where $\bar{\mathcal{R}}(s, a)$ is the expected immediate reward for the state-action pair $(s, a)$.

**Wasserstein Distance.** Let $\mu, \nu \in \mathcal{P}(\mathcal{X})$ be probability measures on a complete and separable metric (Polish) space $(\mathcal{X}, d_\mathcal{X})$. The p-Wasserstein distance between $\mu$ and $\nu$ is (Villani, 2009)

$$\mathcal{W}_p(\mu, \nu) \triangleq \left( \min_{\pi \in \Pi(\mu, \nu)} \int_{\mathcal{X} \times \mathcal{X}} c(x, x') \, d\pi(x, x') \right)^{1/p}, \tag{2}$$

where the cost function is given by the metric as $c(x, x') = (d_\mathcal{X}(x, x'))^p$ for some $p \geq 1$. $\Pi(\mu, \nu)$ is a set of all admissible transport plans between $\mu$ and $\nu$, i.e. probability measures on $\mathcal{X} \times \mathcal{X}$ space with marginals $\mu$ and $\nu$. Wasserstein distances induce geodesic in well-behaved spaces of probability measures. For more discussion, we refer to Appendix A.9. For this work, we consider 1-Wasserstein distance, i.e. $p = 1$, though the results are generalizable to $p > 1$.

**MDPs with Lipschitz Rewards.** Following Pirotta et al. (2015) and Kallel et al. (2024), we assume an MDP with $L_\mathcal{R}$-Lipschitz rewards (ref. Appendix A.1 for elaboration) that satisfies $|\bar{\mathcal{R}}(s, a) - \bar{\mathcal{R}}(s', a')| \leq L_\mathcal{R} d_{\mathcal{S}\mathcal{A}}((s, a), (s', a'))$ for all $s, s' \in \mathcal{S}$ and $a, a' \in \mathcal{A}$. Here, $d_{\mathcal{S}\mathcal{A}}((s, a), (s', a')) = d_\mathcal{S}((s, s')) + d_\mathcal{A}((a, a'))$ is the metric defined on the joint state-action space $\mathcal{S} \times \mathcal{A}$. This is a weaker condition than assuming a completely Lipschitz MDP. Pirotta et al. (2015) showed that for any pair of stationary policies $\pi$ and $\pi'$, the absolute difference between their corresponding objectives is

$$\left| J_\mu^\pi - J_\mu^{\pi'} \right| \leq \frac{L_\mathcal{R}}{\rho} \mathcal{W}_1(v_\pi, v_{\pi'}), \tag{3}$$

where $\mathcal{W}_1(v_\pi, v_{\pi'})$ is the 1-Wasserstein distance between the occupancy measures of the policies (ref. Appendix A.2 for details).

## 3 RL Algorithms as Trajectories of Occupancy Measures

The exploration process (i.e. the exploration-exploitation interplay and learning strategy) of an RL algorithm, influence how the policy model updates its policies (Kaelbling et al., 1996; Sutton & Barto, 2018). During training, a *policy trajectory*, i.e. sequence of policies $(\pi_0, \pi_1, \ldots, \pi_N)$, is generated during policy updates due to the exploratory process. We assume these policies belong to a set of stationary Markov policies parameterised by $\theta$. For policies in this set $\pi_\theta \in \boldsymbol{\Gamma}_\theta$, we define the space of occupancy measures corresponding to $\boldsymbol{\Gamma}_\theta$ as $\mathcal{M} = \{v_{\pi_\theta}(s, a) \mid \pi_\theta \in \boldsymbol{\Gamma}_\theta, \theta \in \mathbb{R}^{N_\theta}\}$.

**Proposition 1** (Properties of $\mathcal{M}$). *If the policy $\pi$ has a smooth parameterization $\theta$ and the inverse of the transition matrix $\boldsymbol{P}^\pi$ exists, then the space of occupancy measures $\mathcal{M}$ is a differentiable manifold. (Proof in Appendix A.3)*

We can endow the manifold $\mathcal{M}$ with a 1-Wasserstein metric $\mathcal{W}_1$ to the compute the length of any path on $\mathcal{M}$ since $(\mathcal{M}, \mathcal{W}_1)$ is a geodesic space (ref. Appendix A.9 for details). The path distance between occupancy measures corresponding to policies parameterized by $\theta, \theta + d\theta \in \mathcal{M}$ is $ds = \mathcal{W}_1(v_{\pi_\theta}, v_{\pi_{\theta+d\theta}})$. Additionally, in imitation learning, the 1-Wasserstein distance between the occupancy measures of the learner and expert can be used as a minimizable loss function to learn the expert's policy (Zhang et al., 2020). Hence, the 1-Wasserstein distance reflects the effort required to achieve this imitation learning. Similarly, we propose the following quantification for the effort to update from one policy to another.

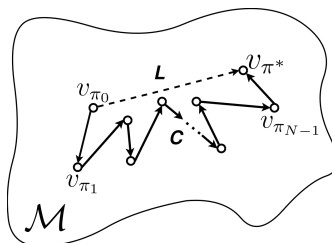

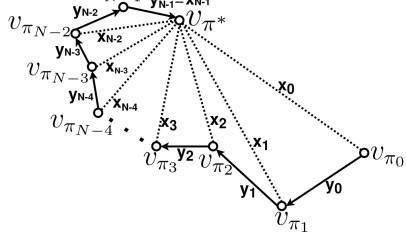

Figure 1: Schematic of the policy trajectory $C$ in the space of occupancy measures $\mathcal{M}$ during RL training (solid line) vs. the geodesic $L$ (shortest path, dashed line) between the initial and final points (i.e. $\pi_0$ and $\pi_N = \pi^*$).

Figure 2: Schematic of how *distance-to-optimal* (denoted by $x_k$) and *stepwise-distance* (denoted by $y_k$) on the occupancy measure space describe exploratory process of an RL algorithm during training.

**Definition 1** (Effort of Learning). *We define the 1-Wasserstein metric between occupancy measures of two policies $\pi$ and $\pi'$, i.e. $\mathcal{W}_1(v_\pi, v_{\pi'})$, as the effort required to learn or update from one policy to the other.*

When a learning process causes an update between occupancy measures in $\mathcal{M}$, we attribute the resulting update effort to the learning process and refer to it as the effort of learning. In a learning process, first the initial policy $\pi_0$ is obtained typically by randomly sampling the model parameters, then these parameters $\theta$ undergo updates until a predefined convergence criterion is satisfied, yielding the final optimal policy $\pi_N = \pi^*$. Since each policy has a corresponding occupancy measure, this process yields a sequence of points on $\mathcal{M}$, which can be connected by geodesics between successive points, producing a curve. The length of the curve is computed by the summation of the finite geodesic distances between consecutive policies along it (Lott, 2008),

$$C \triangleq \sum_{k=0}^{N-1} \mathcal{W}_1(v_{\pi_{\theta_k}}, v_{\pi_{\theta_{k+1}}}), \tag{4}$$

where $\theta_0$ and $\theta_N$ are respectively the initial and final parameter values before and after learning.

## 3.1 EFFORT OF SEQUENTIAL LEARNING (ESL)

As we saw above, RL generates a trajectory in the occupancy measure manifold $\mathcal{M}$, whose length is given by Equation (4). Compared to the long trajectory of sequential policies generated by the exploratory process, the geodesic $L$ is the ideal shortest path to the optimal policy $\pi_N = \pi^*$ from $\pi_0$, whose length is $L = \mathcal{W}_1(v_{\pi_0}, v_{\pi_N})$. This path would be taken by an imitation-learning oracle algorithm that knows $\pi^*$. Both these paths are schematically depicted in Figure 1.

**Definition 2** (Effort of Sequential Learning (ESL)). *We define the effort of sequential learning incurred by a trajectory of the exploratory process of an RL algorithm, relative to the oracle that knows $\pi^*(= \pi_N)$, as*

$$\eta \triangleq \frac{\sum_{k=0}^{N-1} \mathcal{W}_1(v_{\pi_k}, v_{\pi_{k+1}})}{\mathcal{W}_1(v_{\pi_0}, v_{\pi_N})} \tag{5}$$

*Due to the stochasticity of the exploratory process, we introduce an expectation to obtain $\bar{\eta} = \mathbb{E}_{\pi_0, \mu}[\eta]$. We refer to $\bar{\eta}$ as the effort of sequential learning (ESL).*

$\bar{\eta} \geq 1$ and a larger $\bar{\eta}$ correspond to a less efficient exploratory process of the RL algorithm. Hence, an RL algorithm with $\bar{\eta} \approx 1$ closely mimics the oracle and has an efficient exploratory process.

## 3.2 OPTIMAL MOVEMENT RATIO (OMR)

Regret measures the total deviation in value functions incurred by a sequence of policies learned by an RL algorithm with respect to the optimal algorithm that always uses the optimal policy (Sinclair et al., 2023). We show that regret is connected to the sum of distances from each policy in the sequence learned by an RL algorithm to the optimal policy in the occupancy measure space.

**Proposition 2** (Regret and Occupancy Measures). *Given an MDP with $L_{\mathcal{R}}$-Lipschitz rewards, we obtain Regret* $\triangleq \sum_{k=1}^{N} \left( J_{\mu}^{\pi^*} - J_{\mu}^{\pi_k} \right) \leq \frac{L_{\mathcal{R}}}{\rho} \sum_{k=1}^{N} \mathcal{W}_1(v_{\pi_k}, v_{\pi^*})$. *(Proof in Appendix A.4)*

We refer to $\mathcal{W}_1(v_{\pi_k}, v_{\pi^*})$ as the *distance-to-optimal*, and analogously use it as the expected immediate regret in the occupancy measure space. Furthermore, we refer to $\mathcal{W}_1(v_{\pi_k}, v_{\pi_{k+1}})$ as *stepwise-distance*. Interestingly, during training, the *distance-to-optimal* and *stepwise-distance* share a relationship illustrated in Figure 2. From Figure 2, we observe that if the change in *distance-to-optimal*, $\delta_k \triangleq \mathcal{W}_1(v_{\pi_k}, v_{\pi^*}) - \mathcal{W}_1(v_{\pi_{k+1}}, v_{\pi^*}) > 0$, it indicates that the agent got closer to the optimal. We define the set $K^+$ as containing indices $k$ for which $\delta_k > 0$, while $K^-$ contains the rest.

**Definition 3** (Optimal Movement Ratio (OMR)). *We define the proportion of policy transitions that effectively reduce the distance-to-optimal, in a learning trajectory, as*

$$\kappa \triangleq \frac{\sum_{k \in K^+} \mathcal{W}_1(v_{\pi_k}, v_{\pi_{k+1}})}{\sum_{k=0}^{N-1} \mathcal{W}_1(v_{\pi_k}, v_{\pi_{k+1}})} . \tag{6}$$

*Due to the stochasticity of the exploratory process, we introduce an expectation to obtain $\bar{\kappa} = \mathbb{E}_{\pi_0, \mu}[\kappa]$. We refer to $\bar{\kappa}$ as the optimal movement ratio (OMR).*

Note that $\bar{\kappa} \in [0, 1]$, and $\bar{\kappa} \to 1$ indicates that nearly all the policy updates reduce the *distance-to-optimal*, thus showing high efficiency. $\bar{\kappa} \to 0$ implies low efficiency, since only a small fraction of the policy updates contribute towards the reduction of the *distance-to-optimal*.

### 3.3 EXTENSION TO FINITE-HORIZON EPISODIC SETTING

In the episodic finite-horizon MDP formulation of RL, in short *Episodic RL* (Osband et al., 2013; Azar et al., 2017; Ouhamma et al., 2023), the agent interacts with the environment in multiple episodes of $H$ steps. An episode starts by observing state $s_1$. Then, for $t = 1, \ldots H$, the agent draws action $a_t$ from a (possibly time-dependent) policy $\pi_t(\cdot \mid s_t)$, observes the reward $r(s_t, a_t)$, and transits to a state $s_{t+1} \sim T(\cdot \mid s_t, a_t)$. Here, the value function and the state-action value functions at step $h \in [H]$ are defined as $V_h^\pi(s) \triangleq \mathbb{E}_{\mathbb{M}, \pi}\left[ \sum_{t=h}^{H} r(s_t, a_t) \mid s_h = s \right]$, and $Q_h^\pi(s, a) \triangleq \mathbb{E}_{\mathbb{M}, \pi}\left[ \sum_{t=h}^{H} r(s_t, a_t) \mid s_h = s, a_h = a \right]$. Following (Altman, 1999), we can define a finite-horizon version of occupancy measures as

$$v_\pi^H(s, a) \triangleq \frac{1}{H} \sum_{t=1}^{H} \mathbb{P}(s_t = s, a_t = a \mid \pi, \mu). \tag{7}$$

Following (Syed et al., 2008), we can show that $v_\pi^H$ satisfies the linear programming description of value function maximization along with the Bellman flow constraints (ref. Sec II.C. in Kalagarla et al. (2021)). Additionally, we prove that under some assumptions, the finite-horizon occupancy measures also construct a manifold, referred as $\mathcal{M}^H$.

**Proposition 3** (Properties of $\mathcal{M}^H$). *If the policy $\pi$ has a smooth parametrization $\theta$ and the inverses of both the transition matrix $\boldsymbol{P}^\pi$ and $(\mathbb{I} - \boldsymbol{P}^\pi)$ exist, then the space of finite-horizon occupancy measures $\mathcal{M}^H$ is a differentiable manifold. (Proof in Appendix A.5)*

This allows us to similarly define a Wasserstein metric on this manifold, which in turn, allows us to compute ESL and OMR to evaluate different RL algorithms.

## 4 COMPUTATIONAL CHALLENGES AND SOLUTIONS

Similar to regret, our method requires knowing the optimal policy. This is because the efficiency and effectiveness of exploratory processes of RL algorithms are highly coupled with their ability to reach optimality. ESL and OMR depend on the policies being stationary and Markovian.

### 4.1 POLICY DATASETS FOR COMPUTING OCCUPANCY MEASURES

We consider approximations of occupancy measures using datasets assumed to be drawn from these measures. We estimate the Wasserstein distance between the occupancy measures using a method introduced by Alvarez-Melis & Fusi (2020) known as the *optimal transport dataset distance* (OTDD).

OTDD uses datasets to estimate the Wasserstein distance between the underlying distributions. See Appendix A.6 for a detailed account on OTDD.

**Definition 4** (Policy dataset). *A dataset of a policy $\mathcal{D}_\pi$ is a set of state-action pairs drawn from the policy's occupancy measure, i.e. $\mathcal{D}_\pi = \{(s_{(i)}, a_{(i)})\}_{i=1}^m \sim v_\pi$. These can be constituted from the rollouts generated by the policy during task execution.*

We know from imitation learning that if we are given $\mathcal{D}_\pi$, generated by an expert policy, we can train a policy model on it in a supervised manner via behaviour cloning (Hussein et al., 2017). Thus, knowing $\mathcal{D}_\pi$ can allow converting an RL task into a Supervised Learning (SL) task. Consider a scenario when we have access to a sequence of datasets $(\mathcal{D}_{\pi_0}, \ldots, \mathcal{D}_{\pi_N})$, each corresponding to policy $\pi_t$ for $t \geq 0$. If we train (in a supervised manner) a policy model sequentially on these datasets, the model will undergo a similar policy evolution as via the RL algorithm that generated the policy trajectory $(\pi_t)_{t \geq 0}$. This allows us to conceptualise learning in RL as a sequence of SL tasks with sequential transfer learning across the datasets $(\mathcal{D}_{\pi_0}, \ldots, \mathcal{D}_{\pi_N})$. We employ OTDD to estimate $\mathcal{W}_1(v_{\pi_k}, v_{\pi_{k+1}})$ using these datasets, i.e. $d_{OT}(\mathcal{D}_{\pi_k}, \mathcal{D}_{\pi_{k+1}}) \approx \mathcal{W}_1(v_{\pi_k}, v_{\pi_{k+1}})$, based on Proposition 4.

**Proposition 4** (Upper Bound on Estimation Error). *Let an RL algorithm yield a sequence of policies $\pi_0, \ldots, \pi_N$ while training. Now, we construct $N$ datasets $\mathcal{D}_{\pi_0}, \ldots, \mathcal{D}_{\pi_N}$, each consisting of $M$ rollouts of the corresponding policies. Then, we can use these datasets to approximate $\sum_{k=0}^{N-1} \mathcal{W}_1(v_{\pi_k}, v_{\pi_{k+1}})$ by $\sum_{k=0}^{N-1} d_{OT}(\mathcal{D}_{\pi_k}, \mathcal{D}_{\pi_{k+1}})$ with an expected error upper bound $\frac{2N\mathcal{E}_2}{\sqrt{M}} + N\gamma^{T+1} diam(\mathcal{SA})$. Here, $T$ is the total number of steps per episode, $diam(\mathcal{SA})$ is the diameter of the state-action space, and $\mathcal{E}_2$ is a positive-valued and polylogarithmic function of $S$ and $A$. For finite horizon case, we can further reduce the error bound to $\frac{2N\mathcal{E}_2}{\sqrt{M}}$.*

Proof of Proposition 4 is in Appendix A.7. The results support that ESL and OMR can be estimated as

$$\bar{\eta} = \mathbb{E}_{\pi_0, \mu} \left[ \frac{\sum_{k=0}^{N-1} d_{OT}(\mathcal{D}_{\pi_k}, \mathcal{D}_{\pi_{k+1}})}{d_{OT}(\mathcal{D}_{\pi_0}, \mathcal{D}_{\pi_N})} \right], \text{ and } \bar{\kappa} = \mathbb{E}_{\pi_0, \mu} \left[ \frac{\sum_{k \in K^+} d_{OT}(\mathcal{D}_{\pi_k}, \mathcal{D}_{\pi_{k+1}})}{\sum_{k=0}^{N-1} d_{OT}(\mathcal{D}_{\pi_k}, \mathcal{D}_{\pi_{k+1}})} \right]. \quad (8)$$

### 4.2 WHEN AN OPTIMAL POLICY IS NOT REACHED

So far we have assumed that the algorithms converge at the optimal policy, i.e. $\pi_N = \pi^*$. However, this is not always true. We consider a scenario when $\pi_N \neq \pi^*$, and define

$$\eta_{sub} = \frac{\sum_{k=0}^{N-1} \mathcal{W}_1(v_{\pi_k}, v_{\pi_{k+1}})}{\mathcal{W}_1(v_{\pi_0}, v_{\pi_N})}, \pi_N \neq \pi^*. \quad (9)$$

**Proposition 5.** *Given $N \geq 2$ and $\pi_0 \neq \pi_N \neq \pi^*$, we obtain*

$$\frac{\eta - \eta_{sub}}{\eta} \leq \frac{2\mathcal{W}_1(v_{\pi_N}, v_{\pi^*})}{\mathcal{W}_1(v_{\pi_0}, v_{\pi_N})}. \quad (10)$$

This is true due to the triangle inequalities: $\mathcal{W}_1(v_{\pi_0}, v_{\pi^*}) + \mathcal{W}_1(v_{\pi_N}, v_{\pi^*}) \geq \mathcal{W}_1(v_{\pi_0}, v_{\pi_N})$ and $\mathcal{W}_1(v_{\pi_{N-1}}, v_{\pi_N}) + \mathcal{W}_1(v_{\pi_N}, v_{\pi^*}) \geq \mathcal{W}_1(v_{\pi_{N-1}}, v_{\pi^*})$. Equation (10) shows that in the case where $\pi_N$ is close to $\pi^*$, then $\eta_{sub}$ is a good approximation of $\eta$, and thus, a good quantifier to determine the efficiency of the algorithm's exploratory process. The proof is in Appendix A.8 and corresponding experimental results are in Appendix B.5.

## 5 EXPERIMENTAL EVALUATION

In this section, we evaluate the proposed methods in the *2D-Gridworld* and *Mountain Car* (Moore, 1990; Brockman et al., 2016) environments, to analyse our methods in discrete and continuous state-action spaces respectively. The 2D-Gridworld environment is of size 5x5 with actions: {up, right, down, left}. In the gridworld, we perform experiments on 3 settings namely:- A) deterministic with dense rewards, B) deterministic with sparse rewards, and C) stochastic with dense rewards. Further details about these settings are provided in Appendix B.1. The Mountain Car environment, in our experimentation, is a deterministic MDP with dense rewards that consists of both continuous states

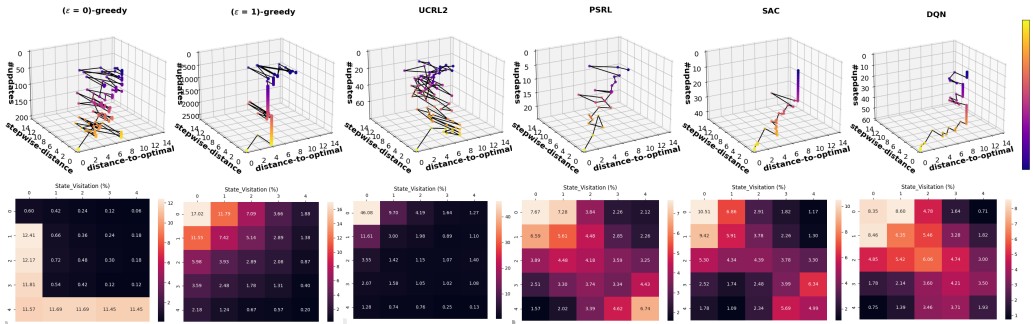

Figure 3: Top row: 3D scatter plots of *distance-to-optimal* (x-axis) and *stepwise-distance* (y-axis) across number of updates (z-axis), illustrating policy evolution in the occupancy measure space for algorithms: $\epsilon(=0)$-greedy and $\epsilon(=1)$-greedy Q-learning, UCRL2, PSRL, SAC, and DQN (left to right). Bottom row: Corresponding state visitation frequencies over the full training. The problem setting is deterministic with dense-rewards and 15 maximum number of steps per episode. (Larger 3D versions and individual 2D projections of these plots are in Appendix D.1)

and actions (described in detail in (Brockman et al., 2016)). The final experiment studies how ESL scales with task hardness in several gridworld environments of varying difficulty.

Our experiments aim to address the following questions:
1. *What information can the visualization of the policy evolution during RL training provide about the exploratory process of the algorithm?*
2. *How do ESL and OMR allow us to analyse the exploratory processes of RL algorithms?*
3. *Does ESL scale proportionally with task difficulty?*

**Summary of Results.** In Section 5.1, we demonstrate that visualizing evolution of *distance-to-optimal* and *stepwise-distance* of different RL algorithms during training reveals: 1) whether the agent is stuck in suboptimal policies, 2) the coverage area of the exploration processes, and 3) their varied characteristics over time. We further compare ESL and OMR of different algorithms on a few environments in Section 5.2. Finally, we show in Section 5.3 that ESL scales proportionally with task difficulty, and thus, reflects the effects of task difficulty on exploration and learning.

## 5.1 EXPLORATION TRAJECTORIES OF RL ALGORITHMS

(I) DISCRETE MDP. To understand the utility of visualizing exploratory processes, we use the following RL algorithms: 1) Tabular Q-learning with a) $\epsilon$-greedy ($\epsilon = 0$) and b) $\epsilon$-greedy ($\epsilon = 1$) strategies; 2) UCRL2 (Jaksch et al., 2010); 3) PSRL (Osband et al., 2013); 4) SAC (Haarnoja et al., 2018; Christodoulou, 2019); and 5) DQN (Mnih et al., 2013) with $\epsilon$-decay. The algorithms solve a simple 5x5 gridworld with dense rewards, starting from top-left (0,0) to reach bottom-right (4,4). Figure 3 presents exploratory behavior of the algorithms in occupancy measure space and state space.

**Q-learning: $\epsilon = 0$ vs $\epsilon = 1$.** Note that $\epsilon = 0$ updates the Q-table by only exploiting, while $\epsilon = 1$ by exploring. From the state visitations, we observe expected characteristics, like a preferred visit path for $\epsilon = 0$ versus $\epsilon = 1$ with visitation frequencies that are similar at states equidistant from the start-state and gradually decreasing as the distance from the state-state increases. From the policy evolution, we see how scattered and erratic the policy transitions are for $\epsilon = 0$. Whereas $\epsilon = 1$ is dominated by unchanging or little-changing policies seen by straight vertical line segments (indicating being 'stuck in suboptimality'). In this setting, $\epsilon = 0$ is characterized by transitioning between diverse policies (i.e. being aggressive with larger coverage area) while $\epsilon = 1$ is likely to be *stuck in suboptimality*. This *stuck in suboptimality* is due to high action randomness in $\epsilon = 1$ that cause the agent to select suboptimal actions, slowing the Q-table convergence and unchanging the learning policy until the best actions are discovered.

**UCRL2 vs PSRL.** UCRL2 has nearly uniform state visits (with the exception of the start-state because the initial state distribution is 1 at state (0,0)), thus being consistent with literature since the algorithm selects exploratory state-action pairs more uniformly (Jaksch et al., 2010). In contrast, PSRL has high visit frequencies along the diagonal states, because it selects actions according to the probability that they are optimal (Osband et al., 2013). We observe from the policy evolution plots that PSRL has smoother policy transitions that are orientated towards optimality, while UCRL2 behaves more aggressively with policy transitions that do not taper as it approaches optimality.

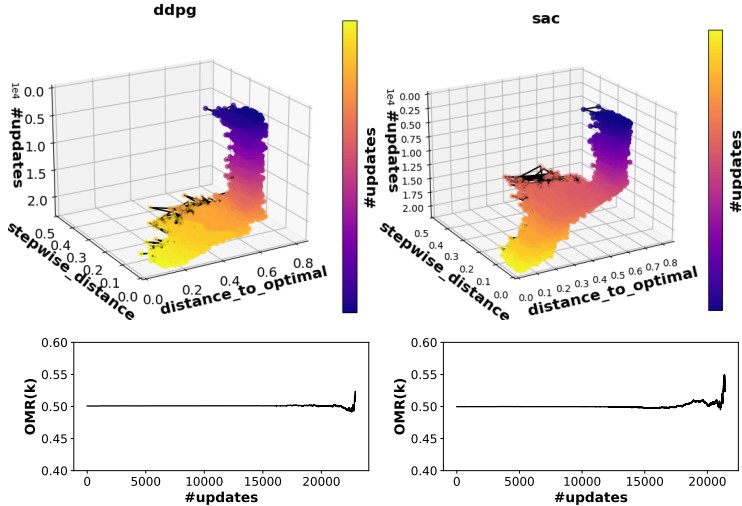

Figure 4: Top row: 3D scatter plots of *distance-to-optimal* and *stepwise-distance* vs. number of updates for DDPG and SAC. Bottom row: OMR($k$) vs. #update, $k$, for the corresponding algorithms.

| Algo. | ESL | OMR | UC | SR% |
|---|---|---|---|---|
| SAC | 9.26±5.54 | 0.58±0.14 | 980±670 | 100 |
| UCRL2 | 47.2±8.20* | 0.49±0.04 | 60.7±11 | 100 |
| PSRL | 23.2±11.5 | 0.52±0.06 | **34.1±9.34** | 100 |
| DQN | 12.4±7.13 | 0.54±0.11 | 161±93 | 98 |
| $\epsilon$(=1)-greedy | **6.27±2.22** | **0.61±0.09** | 672±385 | 100 |
| $\epsilon$(=0.9)-decay | 8.10±3.43 | 0.61±0.10 | 389±138 | 100 |
| $\epsilon$(=0)-greedy | 15.5±5.28 | 0.53±0.06 | 176±37.9 | 84 |

Table 1: Evaluation of RL algorithms (over 40 runs) in the **deterministic, dense-rewards setting** for 5x5 gridworld, including Effort of Sequential Learning (ESL), Optimal Movement Ratio (OMR), number of updates to convergence (UC), and success rate (SR). Lowest ESL, highest OMR and lowest UC values are in **bold**, while the highest ESL value is starred ($\star$).

| Algo. | ESL | OMR | UC | SR% |
|---|---|---|---|---|
| | **Deterministic, sparse** | | | |
| SAC | **27.8±21.9** | **0.57±0.13** | 4385±3274 | 100 |
| UCRL2 | 73.3±0.0 | 0.45±0.0 | **93.0±0.0** | 100 |
| PSRL | 73.2±54.1 | 0.52±0.076 | 100±67.3 | 100 |
| DQN | 137±154* | 0.49±0.08 | 12638±4431 | 80 |
| | **Stochastic, dense** | | | |
| SAC | 445±245 | 0.501±0.004 | 2463±2043 | **92** |
| UCRL2 | 198±121 | 0.502±0.027 | 268±155 | 32 |
| PSRL | **55.4±33.6** | **0.52±0.04** | **76.1±50.6** | **92** |
| DQN | 458±311* | 0.502±0.01 | 1586±1077 | 24 |

Table 2: Evaluation of RL algorithms (over 40 runs) in the **deterministic, sparse-rewards** and **stochastic, dense-rewards** settings for 5x5 gridworld. Lowest ESL, highest OMR and lowest UC values are in **bold**. The highest ESLs are starred.

Osband et al. (2013) highlighted that exploration in PSRL is guided by the variance of sampled policies as opposed to optimism in UCRL2. We observe in Figure 3 that the guiding variance in PSRL reduces after every policy update until optimality is reached, while UCRL2 maintains high variance. *These insights are not reflected by regret as both UCRL2 and PSRL achieve same order of regrets. This shows complementarity of insights yielded by ESL and OMR w.r.t. regret.*

**SAC vs DQN.** The state visits of both the algorithms appear to be similar. SAC has higher visitation frequencies at the corners than DQN. Surprisingly from the policy evolution plots, we learn that both algorithms have a reluctance to transition between policies - hence the *stuck in suboptimality* vertical line segments, especially initially (plotted after removing the filling time for the transitions buffer). This reluctance is due to the slow "soft updates" of target networks (Lillicrap et al., 2016) in the algorithms. We also observe that SAC approaches optimality more gradually than DQN.

**All algorithms.** Figure 3 shows that UCRL2 was more meandering (with larger coverage area) towards optimality than the rest. SAC and DQN approached optimality more directly and smoothly (with smaller coverage area) than the rest. These characteristics are intuitively revealed by policy visualization plots, and are aligned with literature, hence enhancing our understanding of the exploratory processes.

| Algo. | ESL | OMR | UC | SR% |
|---|---|---|---|---|
| DDPG | 1881±500 | 0.501 | 23500±5268 | 100 |
| SAC | 1619±189 | 0.5 | 22700±2971 | 100 |

Table 3: Evaluation of RL algorithms in the Mountain Car continuous MDP (over 5 runs). The variances for OMR and UC are negligible.

(II) CONTINUOUS MDP. We use DDPG (Lillicrap et al., 2016) and SAC (Haarnoja et al., 2018) to solve the Mountain Car. The policy evolutions of these algorithms are presented in Figure 4.

**DDPG vs SAC.** Both exhibit short-distances ($< 1$) between policy updates (i.e. small coverage area). They depict no sign of being stuck or settling early on any particular policy, which shows

their continuously exploratory nature. While they begin with almost constant mean *distances-to-optimal* and *stepwise-distances*, SAC drops its mean *distance-to-optimal* earlier than DDPG.

Figure 4 illustrates how OMR changes with update number $k$. OMR($k$) represents OMR starting with the $k^{\text{th}}$ policy as the initial policy, while OMR starts from the $0^{\text{th}}$ policy. Details of computing OMR($k$) are in Appendix B.2. For both algorithms, OMR($k$) remains near chance level ($\sim 0.5$) initially, then sharply increases near the final updates. This suggests that early policy updates are purely exploratory and oblivious to policy improvement but align with the optimal policy just before convergence. The algorithm's efficiency depends on how early this transition occurs, e.g. starting earlier for SAC than DDPG, rendering SAC more efficient.

## 5.2 Comparison of ESL and OMR across RL Algorithms and Environments, and their complementarity to number of updates (UC) and regret

Tables 1-3 showcase how ESL and OMR are summary metrics of the policy trajectories during learning by evaluating the algorithms in various settings.

**Dense Rewards.** We observe, in Table 1, that PSRL took the lowest number of updates (UC) to reach the optimal policy in contrast with SAC. Yet, PSRL was meandering more than SAC. The relative directness of SAC is captured by lower ESL and higher OMR compared to PSRL. Even though SAC has larger UC than PSRL, it took a shorter path to optimality than PRSL. This shows that the UC does not necessary correlate with ESL and OMR, and it provides incomplete information about the exploratory processes. Indeed, two algorithms may have the same UC, but different ESL or OMR due to different step-wise distances and varied movement towards optimality.

**Sparse Rewards.** In the sparse rewards setting (Table 2), low performance of DQN is observed in both our metrics and UC. However, SAC is more efficient with lowest ESL and highest OMR, yet UCRL2 has the lowest number of updates (UC). Thus, our metrics complement UC. *UCRL2 is provably regret-optimal, while SAC does not have such rigorous theoretical guarantees but is known to be practically efficient, and this is well captured by ESL and OMR.*

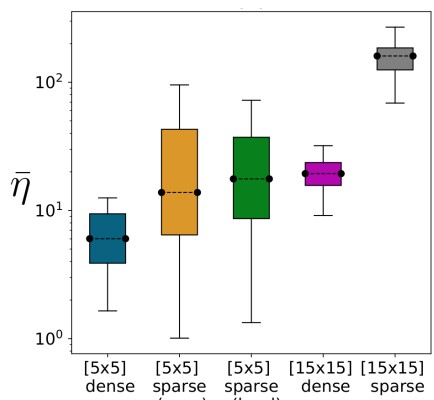

Figure 5: Q-learning with $\epsilon$-greedy ($\epsilon$ = 0.9 decaying, averaged over 40 runs) across deterministic 2D-Gridworld (5x5 and 15x15) tasks. The 1st and 4th (from left to right) have dense rewards, while the rest have sparse rewards (details in Appendix B.1).

**Stochastic Transitions.** In the stochastic setting (Table 2), by observing only successful cases, we notice that the meandering characteristic of PSRL and UCRL2 is more suitable for this setting than SAC and DQN (based on better ESL and OMR values). PSRL and UCRL2 have similar regret bounds (Osband et al., 2013), yet in both Tables 1 and 2, PSRL has better ESL and OMR (along with higher success rate). Thus, our metrics are complementary to regret as well.

Table 3 corroborates with policy evolution plots in Figure 4, in that due to SAC dropping its mean *distance-to-optimal* earlier than DDPG it exhibits a lower ESL. Additionally, we notice a trend of increasing ESL and decreasing OMR across algorithms when shifting from dense-rewards to sparse-rewards settings, from deterministic to stochastic transitions, from discrete to continuous environments, indicating an increase in the effort of the exploratory processes. We have shown how ESL and OMR metrics summarize policy trajectories of algorithms, and that they are complementary to UC and regret. Appendix B.7 highlights further usefulness of these metrics.

## 5.3 ESL Increases with Task Difficulty

Figure 5 illustrates the ESLs for Q-learning with $\epsilon$-decay strategy (for $\epsilon$ = 0.9) across tasks with varying hardness. These tasks are deterministic 2D-Gridworld of sizes 5x5 and 15x15 matched with either dense or sparse rewards (as specified in Appendix B.1). We chose to assess the $\epsilon$-decay Q-learning algorithm because it is simple and yet completes all these tasks. We observe that the ESL is lowest for *[5x5] dense* (5x5 grid, dense rewards) and highest for *[15x15] sparse* (15x15 grid, sparse rewards) as anticipated. The results demonstrate that ESL scales proportionally with task difficulty, matching expectations that more difficult tasks demand greater effort of the exploratory process.

## 6 RELATED WORKS

Several prior works have utilized various components leveraged in our work, namely Wasserstein distance, occupancy measures, and the trajectory of RL on a manifold, but for different purposes. Here, we summarise them and elucidate the connections.

In supervised learning, Alvarez-Melis & Fusi (2020) proposed an optimal transport approach, namely Optimal Transport Dataset Distance (OTDD), to quantify the transferability between two supervised learning tasks by computing the similarity (aka distance) between the task datasets. Here, we conceptualise and define the effort of learning for RL, as a sequence of such supervised learning tasks. We observe that *the total effort of sequential learning can be computed as the sum of OTDD distances between consecutive occupancy measures*. Recently, Zhu et al. (2024) have developed generalized occupancy models by defining cumulative features that are transferable across tasks. In future, one can generalize our indices for the cumulative features constructed from some invertible functions of the step-wise occupancy measures.

Optimal transport-based approaches are also explored in RL literature. These works broadly belong to two families. First line of works uses Wasserstein distance over a posterior distribution of Q-values (Metelli et al., 2019; Likmeta et al., 2023) or return distributions (Sun et al., 2022) to quantify uncertainty, and then to use this Wasserstein distance as a loss to learn better models of the posterior distribution of Q-values or return distributions, respectively. The second line of works uses Wasserstein distance between a feasible family of MDPs as an additional robustness constraint to design robust RL algorithms (Abdullah et al., 2019; Derman & Mannor, 2020; Hou et al., 2020). Here, *we bring a novel concept of using Wasserstein distance between occupancy measures to understand the exploratory dynamics*. Incorporating this insight into better algorithm design would be an interesting future work. Recently, Calo et al. (2024) relate Wasserstein distance between reward-labelled Markov chains to bisimulation metrics which abstract state spaces. In the same spirit, we could use reward as the cost-function in computing our nested Wasserstein distance (OTDD) to obtain a reward- or value-aware OTDD to define broader bisimulation metrics with abstract state-action spaces, instead of just state spaces.

As a parallel approach to optimal transport, the information geometries of the trajectory of an RL algorithm under different settings are studied. These approaches use mutual information as a metric instead of Wasserstein distance. Basu et al. (2020) study the information geometry of Bayesian multi-armed bandit algorithms. They consider a bandit algorithm as a trajectory on a belief-reward manifold, and propose a geometric approach to design a near-optimal Bayesian bandit algorithm. Eysenbach et al. (2021); Laskin et al. (2022) study information geometry of unsupervised RL and propose mutual information maximization schemes over a set of tasks and their marginal state distributions. Yang et al. (2024) extend this approach with Wasserstein distance and demonstrate benefits of using Wasserstein distance than mutual information. *We use Wasserstein distance as a natural metric in occupancy manifold that also allows comparison of hardness of different tasks*. It would be interesting to extend our framework to understand the dynamics of unsupervised RL algorithms.

## 7 DISCUSSION AND FUTURE WORKS

Our work introduces methods to theoretically and quantitatively understand and compare the learning strategies of different RL algorithms. Since learning in a typical RL algorithm happens through a sequence of policy updates, we propose to understand the learning process by visualizing and analysing the path traversed by an RL algorithm in the space of occupancy measures corresponding to this sequence.

We show the usefulness of this approach by conducting experiments on various environments. Our results show that the indices ESL and OMR provide insight into the agent's policy evolution, revealing whether it is steadily approaching the optimal policy or mostly meandering. Additionally, this allows us to understand how the learning process of the same algorithm changes with different rewards and transitions structures, and task hardness. A key limitation of our indices is that they are based on assumption that the final policy reached at the end of training is an optimal one, though we could still derive some benefit from our approach even if not (see Appendix B.7). In the future, it would be interesting to use this approach to benchmark and compare the learning dynamics of different RL algorithms on further environments. In addition, it would be useful to study whether the occupancy measures trajectory of an algorithm provides insights to improve its exploratory process.

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

# A THEORETICAL ANALYSIS

## A.1 MDP WITH LIPSCHITZ REWARDS

Given two metric spaces $(\mathcal{X}, d_{\mathcal{X}})$ and $(\mathcal{Y}, d_{\mathcal{Y}})$, a function $f : \mathcal{X} \to \mathcal{Y}$ is called 1-Lipschitz continuous if (Villani, 2009):

$$d_Y(f(x), f(x')) \leq d_X(x, x'), \forall (x, x') \in X \tag{11}$$

This implies that the Lipschitz semi-norm over the function space $\mathcal{F}(X, Y)$, defined as

$$\|f\|_L = \sup_{x \neq x'} \left\{ \frac{d_Y(f(x), f(x'))}{d_X(x, x')} \mid \forall (x, x') \in \mathcal{X} \right\}, \tag{12}$$

is $\leq 1$. When $(\mathcal{X}, d_{\mathcal{X}})$ is a Polish space and $\mu, \nu \in \mathcal{P}(\mathcal{X})$, the **Kantorovich-Rubinstein** formula states that (Villani, 2009):

$$
\begin{aligned}
\mathcal{W}_1(\mu, \nu) &= \sup_{\|f\|_L \leq 1} \left\{ \int_{\mathcal{X}} f \, d\mu - \int_{\mathcal{X}} f \, d\nu \right\} \\
&= \sup_{\|f\|_L \leq 1} \left\{ \mathbb{E}_\mu \left[ f(X) \right] - \mathbb{E}_\nu \left[ f(X) \right] \right\},
\end{aligned}
\tag{13}
$$

where $W_1(\mu, \nu)$ is the 1-Wasserstein distance between $\mu$ and $\nu$ with $f$ as the cost function.

Note that when $\|f\|_L \leq L_{\mathcal{R}}$ for any $L_{\mathcal{R}} > 0$, then function $f$ is called $L_{\mathcal{R}}$-Lipschitz continuous, and Equation (13) becomes (Gelada et al., 2019),

$$\mathcal{W}_1(\mu, \nu) = \frac{1}{L_{\mathcal{R}}} \sup_{\|f\|_L \leq L_{\mathcal{R}}} \left\{ \mathbb{E}_\mu \left[ f(X) \right] - \mathbb{E}_\nu \left[ f(X) \right] \right\}. \tag{14}$$

Now, we consider $\mathcal{X} = \mathcal{S} \times \mathcal{A}$, i.e. the state-action space, $\mathcal{Y} = \mathbb{R}$, i.e. the real line, and the function $f$ to be the reward function $\bar{\mathcal{R}}$. Then, we can call the reward function $\bar{\mathcal{R}}$ to be $L_{\mathcal{R}}$-Lipschitz if

$$|\bar{\mathcal{R}}(s, a) - \bar{\mathcal{R}}(s', a')| \leq L_{\mathcal{R}} d_{\mathcal{S}\mathcal{A}}((s, a), (s', a'))$$

for all $s, s' \in \mathcal{S}$, and $a, a' \in \mathcal{A}$, and $d_{\mathcal{S}\mathcal{A}}((s, a), (s', a')) = d_{\mathcal{S}}((s, s')) + d_{\mathcal{A}}((a, a'))$ being the metric on the state-action space $\mathcal{S} \times \mathcal{A}$. If the reward function $\mathcal{R}$ of an MDP is $L_{\mathcal{R}}$-Lipschitz, we refer it as an MDP with Lipschitz rewards.

## A.2 PERFORMANCE DIFFERENCE AND OCCUPANCY MEASURES

We know that

$$J_\mu^\pi = \frac{1}{\rho} \mathbb{E}_{(s,a) \sim v_\pi} \left[ \bar{\mathcal{R}}(s, a) \right] . \tag{15}$$

Using Equation (15), we write for two policies $\pi$ and $\pi'$, with $\mu(s)$ as the initial state distribution,

$$\left| J_\mu^\pi - J_\mu^{\pi'} \right| = \frac{1}{\rho} \left| \mathbb{E}_{(s,a) \sim v_\pi} \left[ \bar{\mathcal{R}}(s, a) \right] - \mathbb{E}_{(s,a) \sim v_{\pi'}} \left[ \bar{\mathcal{R}}(s, a) \right] \right| \tag{16}$$

Given an MDP with $L_{\mathcal{R}}$-Lipschitz rewards, the **Kantorovich-Rubinstein** formula dictates that (Gelada et al., 2019):

$$\sup_{\|\bar{R}\|_L \leq L_{\mathcal{R}}} \left| \mathbb{E}_{(s,a) \sim v_\pi} \left[ \bar{\mathcal{R}}(s, a) \right] - \mathbb{E}_{(s,a) \sim v_{\pi'}} \left[ \bar{\mathcal{R}}(s, a) \right] \right| = L_{\mathcal{R}} \mathcal{W}_1(v_\pi, v_{\pi'}) \tag{17}$$

By dividing both sides of Equation (17) by $\rho$, and due to an upper bound by the supremum, this inequality follows:

$$\left| J_\mu^\pi - J_\mu^{\pi'} \right| \leq \frac{L_{\mathcal{R}}}{\rho} \mathcal{W}_1(v_\pi, v_{\pi'}) \tag{18}$$

### A.3 PROOF OF PROPOSITION 1

The Linear Programming formulation for solving MDPs, assuming discrete state and action spaces, is (Puterman, 1994):

$$\max_{v_\pi} \sum_{s,a} r(s,a)v_\pi(s,a)$$

$$\text{subject to} \sum_a v_\pi(s,a) = p_0(s) + \gamma \sum_{s',a} T(s \mid s',a)v_\pi(s',a) \tag{19}$$

$$v_\pi(s,a) \geq 0 \quad \forall(s,a) \in \mathcal{S} \times \mathcal{A},$$

where $p_0(s)$ is the initial state distribution and $T(s \mid s',a)$ is the transition probability. The constraints of this optimization problem are often referred to as *Bellman Flow Constraint*.

A stationary policy $\pi$ has a corresponding occupancy measure $v_\pi(s,a)$ that satisfies the Bellman flow constraint (Syed et al., 2008), and hence $\pi$ and $v_\pi(s,a)$ share a bijective relationship (Syed et al., 2008; Givchi, 2021),

$$\pi(a \mid s) = \frac{v_\pi(s,a)}{u_\pi(s)} \tag{20}$$

with

$$u_\pi(s) = \sum_{a'} v_\pi(s,a') = p_0(s) + \gamma \sum_{s',a'} T(s \mid s',a')v_\pi(s',a') \tag{21}$$

By rearranging Equation (20) to

$$v_\pi(s,a) = \pi(a \mid s)u_\pi(s) \tag{22}$$

and substituting Equation (22) into Equation (21), we can rewrite Equation (21) as (defining $\mathcal{P}^\pi \triangleq \sum_a T(s \mid s',a)\pi(a \mid s')$),

$$p_0(s) = u_\pi(s) - \gamma \sum_{s',a} T(s \mid s',a)\pi(a \mid s')u_\pi(s')$$

$$\triangleq u_\pi(s) - \gamma \sum_{s'} \mathcal{P}^\pi(s \mid s')u_\pi(s') \tag{23}$$

which in matrix form is

$$\mathbf{p}_0 = \mathbf{u}_\pi - \gamma \mathbf{P}^\pi \mathbf{u}_\pi$$

$$= (\mathbb{I} - \gamma \mathbf{P}^\pi)\,\mathbf{u}_\pi, \tag{24}$$

where $\mathbf{p}_0, \mathbf{u}_\pi \in \mathbb{R}^{|\mathcal{S}|}$ are column vectors and $\mathbf{P}^\pi \in \mathbb{R}^{|\mathcal{S}| \times |\mathcal{S}|}$ are matrices. Solving for $\mathbf{u}_\pi$, we get

$$\mathbf{u}_\pi = (\mathbb{I} - \gamma \mathbf{P}^\pi)^{-1}\mathbf{p}_0 \tag{25}$$

The inverse matrix $(\mathbb{I} - \gamma \mathbf{P}^\pi)^{-1}$ exists because for $\gamma < 1$, $(\mathbb{I} - \gamma \mathbf{P}^\pi)$ is a strictly diagonally dominant matrix (Syed et al., 2008). Thus, $(\mathbb{I} - \gamma \mathbf{P}^\pi)^{-1} = \sum_{t=0}^{\infty}(\gamma \mathbf{P}^\pi)^t$, where $\sum_{t=0}^{\infty}(\gamma \mathbf{P}^\pi)^t$ forms a valid *Neumann series* (Ward, 2021). We let $\mathbf{A}^\pi = \sum_{t=0}^{\infty}(\gamma \mathbf{P}^\pi)^t$, so Equation (24) can be written as $\mathbf{u}_\pi = \mathbf{A}^\pi \mathbf{p}_0$. We can therefore express Equation (22) in matrix form as:

$$\mathbf{v}_\pi = \mathbf{\Pi} \odot \left(\mathbf{u}_\pi^T \otimes \mathbf{1}\right)^T$$

$$= \mathbf{\Pi} \odot \left(\mathbf{p}_0^T(\mathbf{A}^\pi)^T \otimes \mathbf{1}\right)^T, \tag{26}$$

where $\mathbf{\Pi}, \mathbf{v}_\pi \in \mathbb{R}^{|\mathcal{S}| \times |\mathcal{A}|}$, $\mathbf{1} \in \mathbb{R}^{|\mathcal{A}|}$ is a column vector of ones, $\otimes$ presents the Kronecker product, and $\odot$ denotes the Hadamard product.

If we consider the case of a parameterized policy $\mathbf{\Pi}(\theta)$, then the derivative of $\mathbf{v}_\pi$ with respect to $\theta$ is

$$\nabla_\theta \mathbf{v}_\pi = \nabla_\theta \left[\mathbf{\Pi} \odot \left(\mathbf{p}_0^T(\mathbf{A}^\pi)^T \otimes \mathbf{1}\right)^T\right]$$

$$= \nabla_\theta \mathbf{\Pi} \odot \left(\mathbf{p}_0^T(\mathbf{A}^\pi)^T \otimes \mathbf{1}\right)^T + \mathbf{\Pi} \odot \nabla_\theta \left(\mathbf{p}_0^T(\mathbf{A}^\pi)^T \otimes \mathbf{1}\right)^T \tag{27}$$

$$= \nabla_\theta \mathbf{\Pi} \odot \left(\mathbf{p}_0^T(\mathbf{A}^\pi)^T \otimes \mathbf{1}\right)^T + \mathbf{\Pi} \odot \left(\mathbf{p}_0^T(\nabla_\theta \mathbf{A}^\pi)^T \otimes \mathbf{1}\right)^T$$

The first term in Equation (36) is differentiable since the policy is parameterized by $\theta$. We expand $\nabla_\theta \mathbf{A}^\pi$ as follows:

$$
\begin{aligned}
\nabla_\theta \mathbf{A}^\pi &= \sum_{t=0}^\infty t(\gamma \mathbf{P}^\pi)^{t-1} \gamma \nabla_\theta \mathbf{P}^\pi \\
&\equiv \sum_{t=0}^\infty t(\gamma \mathbf{P}^\pi)^{t-1} \gamma \nabla_\theta \left[ \sum_{s',a} T(s|s',a)\pi(a|s') \right] \\
&= \sum_{t=0}^\infty t(\gamma \mathbf{P}^\pi)^{t-1} \gamma \left[ \sum_{s',a} T(s|s',a)\nabla_\theta \pi(a|s') \right] \\
&= \sum_{t=0}^\infty t(\gamma \mathbf{P}^\pi)^t (\mathbf{P}^\pi)^{-1} \left[ \sum_{s',a} T(s|s',a)\nabla_\theta \pi(a|s') \right]
\end{aligned}
\tag{28}
$$

If $(\mathbf{P}^\pi)^{-1}$ exists, then $\nabla_\theta \mathbf{A}^\pi$ is differentiable, and consequently so is $\nabla_\theta \mathbf{v}_\pi$, based on Equation (36) and Equation (37). Proceeding similarly, given the same conditions, we see that all higher derivatives of $v_\pi$ also exist with respect to $\theta$. Thus, the space of parametrized occupancy measures $v_\pi$ forms a differentiable manifold.

## A.4 PROOF OF PROPOSITION 2

Regret is a common metric for evaluating agents, that measures the total loss an agent incurs over policy updates by using its policy in lieu of the optimal one, defined as (Osband et al., 2013),

$$
\text{Regret} = \mathbb{E}_{s\sim\mu} \left[ \sum_k (V^*(s) - V_{\pi_k}(s)) \right]
\tag{29}
$$

where $V^* = V_{\pi^*}$ is the value function of the optimal policy $\pi^*$ while $V_{\pi_k}(s)$ is the value function of policy $\pi_k$, and $\mu$ is the initial state distribution.

Since $J_\mu^\pi = \mathbb{E}_{s\sim\mu}[V_\pi(s)]$, we can conclude from Equation (29) that

$$
\begin{aligned}
\text{Regret} &= \mathbb{E}_{s\sim\mu} \left[ \sum_k (V^*(s) - V_{\pi_k}(s)) \right] \\
&= \sum_k \left[ \mathbb{E}_{s\sim\mu}(V^*(s) - V_{\pi_k}(s)) \right] \\
&= \sum_k \left( J_\mu^{\pi^*} - J_\mu^{\pi_k} \right) \\
&= \sum_k \left| J_\mu^{\pi^*} - J_\mu^{\pi_k} \right| \\
&\leq \sum_k \frac{L_{\mathcal{R}}}{\rho} \mathcal{W}_1(v_{\pi^*}, v_{\pi_k})
\end{aligned}
\tag{30}
$$

The last inequality is due to Equation (18).

## A.5 PROOF OF PROPOSITION 3

Let us begin the proof by defining the visitation probability at any step $h \in [H]$ in an episode, following policy $\pi(a|s)$. Specifically,

$$
q_\pi^h(s,a) \triangleq \mathbb{P}(s_h = s, a_h = a) \;\; \forall h \in [H] \;\; \text{and} \;\; q_\pi^h(s,a) \triangleq 0 \;\; \forall h \in \mathbb{N} \wedge h > H.
\tag{31}
$$

Thus, we rewrite Equation (7) $v_\pi^H(s,a) = \frac{1}{H} \sum_{h=1}^H q_\pi^h(s,a)$.

Then, following (Kalagarla et al., 2021), we can write the Linear Programming formulation for solving episodic MDP $\mathbb{M}^H$ as

$$
\max_{\{q_\pi^h\}_{h=1}^H} \sum_{h,s,a} r(s,a) q_\pi^h(s,a)
$$

$$
\text{subject to} \sum_a q_\pi^h(s,a) = \sum_{s',a} T(s \mid s',a) q_\pi^{h-1}(s',a) \quad \forall h \in [H] \wedge h > 1\,,
$$

$$
q_\pi^1(s,a) = \pi(a|s)\mu(s)\,,
$$

$$
q_\pi^h(s,a) \geq 0 \qquad \forall h \in [H], (s,a) \in \mathcal{S} \times \mathcal{A}\,,
$$

(32)

where $\mu(s)$ is the initial state distribution and $T(s \mid s',a)$ is the transition probability. The constraints of this optimization problem are often referred to as *Bellman Flow Constraints*.

This implies that

$$
\sum_{h=2}^{H+1} \sum_a q_\pi^h(s,a) = \sum_{h=2}^{H+1} \sum_{s',a} T(s \mid s',a) q_\pi^{h-1}(s',a)
$$

$$
\implies \sum_a q_\pi^1(s,a) + \sum_{h=2}^{H+1} \sum_a q_\pi^h(s,a) = \sum_{h=2}^{H+1} \sum_{s',a} T(s \mid s',a) q_\pi^{h-1}(s',a) + \sum_a q_\pi^1(s,a)
$$

$$
\implies \sum_a \sum_{h=1}^{H+1} q_\pi^h(s,a) = \sum_{h=2}^{H+1} \sum_{s',a} T(s \mid s',a) q_\pi^{h-1}(s',a) + \sum_a q_\pi^1(s,a)
$$

$$
\implies H \sum_a v_\pi^H(s,a) = \sum_{s',a} T(s \mid s',a) \Big( \sum_{h=2}^{H+1} q_\pi^{h-1}(s',a) \Big) + \mu(s)
$$

$$
\implies H \sum_a v_\pi^H(s,a) = H \sum_{s',a} T(s \mid s',a) v_\pi^H(s',a) + \mu(s)
$$

$$
\implies \sum_a v_\pi^H(s,a) = \sum_{s',a} T(s \mid s',a) v_\pi^H(s',a) + \frac{1}{H}\mu(s)
$$

$$
\implies u_\pi^H(s) \triangleq \sum_a v_\pi^H(s,a) = \sum_{s',a} T(s \mid s',a)\pi(a|s') u_\pi^H(s') + \frac{1}{H}\mu(s)\,.
$$

(33)

Now, we denote $\mathbf{u}_\pi^H$ and $\bar{\mu}$ as corresponding column vectors and the transition matrix $\mathbf{P}^\pi \triangleq \Big[ \sum_{s',a} T(s \mid s',a)\pi(a|s') \Big]$. Thus, we obtain

$$
(\mathbb{I} - \mathbf{P}^\pi)\mathbf{u}_\pi^H = \frac{1}{H}\bar{\mu} \implies \mathbf{u}_\pi^H = \frac{1}{H}(\mathbb{I} - \mathbf{P}^\pi)^{-1}\bar{\mu}\,.
$$

(34)

We can therefore express the finite horizon occupancy measure in matrix form as

$$
\mathbf{v}_\pi^H = \mathbf{\Pi} \odot \big( (\mathbf{u}_\pi^H)^T \otimes \mathbf{1} \big)^T = \mathbf{\Pi} \odot \big( \bar{\mu}^T (\mathbf{A}_H^\pi)^T \otimes \mathbf{1} \big)^T
$$

(35)

where $\mathbf{\Pi}, \mathbf{v}_\pi \in \mathbb{R}^{|\mathcal{S}| \times |\mathcal{A}|}$, $\mathbf{1} \in \mathbb{R}^{|\mathcal{A}|}$ is a column vector of ones, $\otimes$ presents the Kronecker product, $\odot$ denotes the Hadamard product, and $\mathbf{A}_H^\pi \triangleq \frac{1}{H}(\mathbb{I} - \mathbf{P}^\pi)^{-1}$.

If we consider the case of a parameterized policy $\mathbf{\Pi}(\theta)$, the derivative of $\mathbf{v}_\pi^H$ with respect to $\theta$ is

$$
\begin{aligned}
\nabla_\theta \mathbf{v}_\pi^H &= \nabla_\theta \Big[ \mathbf{\Pi} \odot \big( \bar{\mu}^T (\mathbf{A}_H^\pi)^T \otimes \mathbf{1} \big)^T \Big] \\
&= \nabla_\theta \mathbf{\Pi} \odot \big( \bar{\mu}^T (\mathbf{A}_H^\pi)^T \otimes \mathbf{1} \big)^T + \mathbf{\Pi} \odot \nabla_\theta \big( \bar{\mu}^T (\mathbf{A}_H^\pi)^T \otimes \mathbf{1} \big)^T \\
&= \nabla_\theta \mathbf{\Pi} \odot \big( \bar{\mu}^T (\mathbf{A}_H^\pi)^T \otimes \mathbf{1} \big)^T + \mathbf{\Pi} \odot \big( \bar{\mu}^T (\nabla_\theta \mathbf{A}_H^\pi)^T \otimes \mathbf{1} \big)^T
\end{aligned}
$$

(36)

The first term in Equation (36) is differentiable since the policy is parameterized by $\theta$. We expand on $\nabla_\theta \mathbf{A}_H^\pi$ as follows:

$$
\begin{aligned}
H \nabla_\theta \mathbf{A}_H^\pi &= \nabla_\theta (\mathbb{I} - \mathbf{P}^\pi)^{-1} \\
&= \nabla_\theta \left( \sum_{i=0}^\infty (\mathbf{P}^\pi)^i \right) \\
&= \sum_{i=0}^\infty i (\mathbf{P}^\pi)^{i-1} \nabla_\theta \mathbf{P}^\pi \\
&= \sum_{i=0}^\infty i (\mathbf{P}^\pi)^{i-1} \left[ \sum_{s',a} T(s|s',a) \nabla_\theta \pi(a|s') \right].
\end{aligned}
\tag{37}
$$

If $(\mathbf{P}^\pi)^{-1}$ exists, then $\nabla_\theta \mathbf{A}_H^\pi$ is differentiable, and consequently so is $\nabla_\theta \mathbf{v}_\pi^H$. Proceeding similarly, given the same conditions, we see that all higher derivatives of $\mathbf{v}_\pi^H$ also exist with respect to $\theta$. Thus, the space of parametrized finite-horizon occupancy measures $v_\pi^H$ forms a differentiable manifold $\mathcal{M}^H$.

### A.6 OPTIMAL TRANSPORT DATASET DISTANCE (OTDD)

Suppose we have two datasets, each consisting of feature-label pairs, $\mathcal{D}_A = \{(t_A^i, u_A^i)\}_{i=1}^m \sim P_A(t, u)$ and $\mathcal{D}_B = \{(t_B^i, u_B^i)\}_{i=1}^n \sim P_B(t, u)$ with $t_A, t_B \in \mathcal{T}$ and $u_A, u_B \in \mathcal{U_A}, \mathcal{U_B}$. These datasets can be used to create empirical distributions $\hat{P}_A(t, u)$ and $\hat{P}_B(t, u)$. OTDD is the p-Wasserstein distance between the datasets $\mathcal{D}_A$ and $\mathcal{D}_B$ - which is essentially the distance between their empirical distributions $\hat{P}_A$ and $\hat{P}_B$ - with the cost function defined as the metric of the joint space $\mathcal{T} \times \mathcal{U}$ (Alvarez-Melis & Fusi, 2020).

Naturally, the metric on this joint space can be defined as $d_{\mathcal{TU}}((t, u), (t', u')) = (d_\mathcal{T}(t, t')^p + d_\mathcal{U}(u, u')^p)^{1/p}$, for $p \geq 1$. However, in most applications $d_\mathcal{T}$ is readily available, while $d_\mathcal{U}$ might be scarce, especially in supervised learning (SL) between labels from unrelated label sets (Alvarez-Melis & Fusi, 2020). Further, we want $d_\mathcal{T}$ and $d_\mathcal{U}$ to have the same units to be addable. To overcome these issues, $d_\mathcal{U}$ is expressed in terms of $d_\mathcal{T}$ by mapping labels $u$ to distributions over the feature space $\mathcal{P}(\mathcal{T})$ as $u \to \alpha_u(T) \triangleq P(T \mid U = u) \in \mathcal{P}(\mathcal{T})$. Therefore, the distance between the labels $u$ and $u'$ is defined as the p-Wasserstein distance between $\alpha_u(T)$ and $\alpha_{u'}(T)$,

$$
\begin{aligned}
d_\mathcal{U}(u, u') &= \mathcal{W}_p^p(\alpha_u(T), \alpha_{u'}(T)) \\
&= \min_{\pi \in \Pi(\alpha_u, \alpha_{u'})} \int_{\mathcal{T} \times \mathcal{T}} (d_\mathcal{T}(t, t'))^p \, d\pi(t, t')
\end{aligned}
\tag{38}
$$

The metric on the joint space becomes,

$$
d_{\mathcal{TU}}((t, u), (t', u')) = \left( d_\mathcal{T}(t, t')^p + \mathcal{W}_p^p(\alpha_u(T), \alpha_{u'}(T)) \right)^{1/p}
\tag{39}
$$

Let $\mathcal{Z} = \mathcal{T} \times \mathcal{U}$, then the p-Wasserstein distance between $\hat{P}_A(t, u)$ and $\hat{P}_B(t, u)$ is a "nested" Wasserstein distance:

$$
\begin{aligned}
\mathcal{W}_p^p(\hat{P}_A, \hat{P}_B) &= \min_{\pi \in \Pi(P_A, P_B)} \int_{\mathcal{Z} \times \mathcal{Z}} (d_\mathcal{Z}(z, z'))^p \, d\pi \\
&= \min_{\pi \in \Pi(P_A, P_B)} \int_{\mathcal{TU} \times \mathcal{TU}} \left( d_\mathcal{T}(t, t')^p + \mathcal{W}_p^p(\alpha_u, \alpha_{u'}) \right) \, d\pi
\end{aligned}
\tag{40}
$$

$W_p^p(\hat{P}_A, \hat{P}_B)$ is the OTDD between datasets $\mathcal{D}_A$ and $\mathcal{D}_B$, often expressed as $d_{OT}(\mathcal{D}_A, \mathcal{D}_B)$. This is used in transfer learning to determine the distance (or similarity) between datasets.

### A.7 PROOF OF PROPOSITION 4

We compute the error in occupancy measure for both the infinite and finite horizon cases. In infinite horizon MDPs, the occupancy measure is defined as the expected discounted number of visits of a state-action pair $(s, a)$ in a trajectory (Laroche & des Combes, 2023): $\mu = (1 - \gamma) \sum_{t=0}^{\infty} \gamma^t \mu_t$, where $\mu_t = P(s_t, a_t \mid \pi, \eta)$ is the state-action probability distribution at time step $t$ with the initial state distribution $\eta$ following the policy $\pi$. In finite horizon MDPs, the occupancy measure is the expected number of visits of a state-action pair $(s, a)$ in an episode of length $H$ (Altman, 1999): $\mu = \frac{1}{H} \sum_{t=1}^{H} \mu_t$.

First, we derive error bounds for the infinite horizon MDP in which $\gamma < 1$ and the occupancy measure is approximated using a finite number of samples collected up to a finite number of time steps $T$. Later, we derive error bounds for the finite horizon MDP.

#### A.7.1 INFINITE HORIZON MDPs

**Estimated Occupancy Measure.** For convenience, we express the occupancy measure as $\mu = (1 - \gamma) \sum_{t=0}^{\infty} \gamma^t \mu_t$, where $\mu_t = P(s_t, a_t \mid \pi, \eta)$ is the state-action probability distribution at time step $t$ with the initial state distribution $\eta$ following the policy $\pi$. To compute $\mu$, we roll out $N$ episodes (each of multiple time steps) using $\pi$, and take $N$ number of samples at $t$ to approximate $\mu_t$. Thus, the empirical occupancy measure $\hat{\mu}$ is given by $\hat{\mu} = \rho \sum_{t=0}^{T} \gamma^t \hat{\mu}_t^N$, where $\rho = \frac{1}{\sum_{t=0}^{T} \gamma^t}$. Note that the total number of samples in the policy dataset $\mathcal{D}_\pi$ is $|\mathcal{D}_\pi| = N(T + 1)$.

**Occupancy Measure Estimation Error.** Consider two occupancy measures $\mu = (1-\gamma) \sum_{t=0}^{\infty} \gamma^t \mu_t$ and $\nu = (1 - \gamma) \sum_{t=0}^{\infty} \gamma^t \nu_t$ (with estimates $\hat{\mu} = \rho \sum_{t=0}^{T} \gamma^t \hat{\mu}_t^{N_\mu}$ and $\hat{\nu} = \rho \sum_{t=0}^{T} \gamma^t \hat{\nu}_t^{N_\nu}$). For independent sets $\{\mu_t\}_{t \geq 0}$ and $\{\nu_t\}_{t \geq 0}$, the Wasserstein distance has the following additive property (Panaretos & Zemel, 2019),

$$\mathcal{W}_p(\sum_t \mu_t, \sum_t \nu_t) \leq \sum_t \mathcal{W}_p(\mu_t, \nu_t) \tag{41}$$

While for $a \in \mathbb{R}$ (Panaretos & Zemel, 2019),

$$\mathcal{W}_p(a\mu, a v) = |a| \mathcal{W}_p(\mu, v) \tag{42}$$

Therefore, for our scenario where $p = 1$, the Wasserstein distance between $\mu$ and $\nu$ is given by:

$$\mathcal{W}_1(\mu, \nu) = \mathcal{W}_1((1 - \gamma) \sum_{t=0}^{\infty} \gamma^t \mu_t, (1 - \gamma) \sum_{t=0}^{\infty} \gamma^t \nu_t)$$
$$\leq (1 - \gamma) \sum_{t=0}^{\infty} \gamma^t \mathcal{W}_1(\mu_t, \nu_t) \tag{43}$$

while for $\hat{\mu}$ and $\hat{\nu}$,

$$\mathcal{W}_1(\hat{\mu}, \hat{\nu}) \leq \rho \sum_{t=0}^{T} \gamma^t \mathcal{W}_1(\hat{\mu}_t^{N_\mu}, \hat{\nu}_t^{N_\nu}) \tag{44}$$

In the RL problems we consider, the state-action space $\mathcal{Z} = \mathcal{S} \times \mathcal{A}$ is commonly defined as the subset of the Euclidean space $\mathcal{Z} \in \mathbb{R}^B$, where usually $B \geq 2$. Theorems 1 and 3 in (Sommerfeld et al., 2019) establish the following error bounds between the true and empirical probability distributions,

$$\mathbb{E}[\mathcal{W}_1(\hat{\mu}_t^{N_\mu}, \mu_t)] \leq \mathcal{E}_2 N_\mu^{-\frac{1}{2}}$$
$$\mathbb{E}[\mathcal{W}_1(\hat{\nu}_t^{N_\nu}, \nu_t)] \leq \mathcal{E}_2 N_\nu^{-\frac{1}{2}} \tag{45}$$

where

$$\mathcal{E}_2 \leq 4B^{1/2} diam(\mathcal{Z}) \cdot \begin{cases} 2 + (1/2)\log_2|\mathcal{Z}| & \text{if } B = 2 \\ |\mathcal{Z}|^{1/2 - 1/B} \left[2 + 1/(2^{B/2-1} - 1)\right] & \text{if } B > 2 \end{cases}$$

Note that $|\mathcal{Z}|$ and $diam(\mathcal{Z})$ denote the cardinality and diameter of $\mathcal{Z}$, respectively.

Suppose $a = \mathcal{W}_1(\hat{\mu}, \hat{\nu})$, $b = \mathcal{W}_1(\hat{\mu}, \mu)$, $c = \mathcal{W}_1(\hat{\nu}, \mu)$, $d = \mathcal{W}_1(\mu, \nu)$, and $e = \mathcal{W}_1(\hat{\nu}, \nu)$. Then by performing two reverse triangle inequalities,

$$
\begin{aligned}
|a - c| \leq b \quad &\text{and} \quad |c - d| \leq e \\
\implies |a - d| &\leq b + e
\end{aligned}
\tag{46}
$$

Equation (46) implies that,

$$
\begin{aligned}
\mathbb{E}[|\mathcal{W}_1(\hat{\mu}, \hat{\nu}) - \mathcal{W}_1(\mu, \nu)|] &\leq \mathbb{E}[\mathcal{W}_1(\hat{\mu}, \mu) + \mathcal{W}_1(\hat{\nu}, \nu)] \\
&= \mathbb{E}[\mathcal{W}_1(\rho \sum_{t=0}^{T} \gamma^t \hat{\mu}_t^{N_\mu}, \mu) + \mathcal{W}_1(\rho \sum_{t=0}^{T} \gamma^t \hat{\nu}_t^{N_\nu}, \nu)] \\
&= \mathbb{E}[\mathcal{W}_1(\rho \sum_{t=0}^{T} \gamma^t \hat{\mu}_t^{N_\mu}, \mu)] + \mathbb{E}[\mathcal{W}_1(\rho \sum_{t=0}^{T} \gamma^t \hat{\nu}_t^{N_\nu}, \nu)] \\
&\quad + \mathbb{E}[\mathcal{W}_1((1-\gamma) \sum_{t=0}^{\infty} \gamma^t \hat{\mu}_t^{N_\mu}, \mu) - \mathcal{W}_1((1-\gamma) \sum_{t=0}^{\infty} \gamma^t \hat{\mu}_t^{N_\mu}, \mu)] \\
&\quad + \mathbb{E}[\mathcal{W}_1((1-\gamma) \sum_{t=0}^{\infty} \gamma^t \hat{\nu}_t^{N_\nu}, \nu) - \mathcal{W}_1((1-\gamma) \sum_{t=0}^{\infty} \gamma^t \hat{\nu}_t^{N_\nu}, \nu)]
\end{aligned}
\tag{47}
$$

By virtue of triangle inequalities, we get

$$
\begin{aligned}
\mathcal{W}_1(\rho \sum_{t=0}^{T} \gamma^t \hat{\mu}_t^{N_\mu}, (1-\gamma) \sum_{t=0}^{\infty} \gamma^t \hat{\mu}_t^{N_\mu}) &\geq \mathcal{W}_1(\rho \sum_{t=0}^{T} \gamma^t \hat{\mu}_t^{N_\mu}, \mu) - \mathcal{W}_1((1-\gamma) \sum_{t=0}^{\infty} \gamma^t \hat{\mu}_t^{N_\mu}, \mu) \\
\mathcal{W}_1(\rho \sum_{t=0}^{T} \gamma^t \hat{\nu}_t^{N_\nu}, (1-\gamma) \sum_{t=0}^{\infty} \gamma^t \hat{\nu}_t^{N_\nu}) &\geq \mathcal{W}_1(\rho \sum_{t=0}^{T} \gamma^t \hat{\nu}_t^{N_\nu}, \nu) - \mathcal{W}_1((1-\gamma) \sum_{t=0}^{\infty} \gamma^t \hat{\nu}_t^{N_\nu}, \nu)
\end{aligned}
\tag{48}
$$

Therefore, the right-hand-side (R.H.S) of Equation (47) can be further simplified as

$$
\begin{aligned}
\text{R.H.S} &\leq \mathbb{E}[\mathcal{W}_1(\rho \sum_{t=0}^{T} \gamma^t \hat{\mu}_t^{N_\mu}, (1-\gamma) \sum_{t=0}^{\infty} \gamma^t \hat{\mu}_t^{N_\mu})] + \mathbb{E}[\mathcal{W}_1(\rho \sum_{t=0}^{T} \gamma^t \hat{\nu}_t^{N_\nu}, (1-\gamma) \sum_{t=0}^{\infty} \gamma^t \hat{\nu}_t^{N_\nu})] \\
&\quad + \mathbb{E}[\mathcal{W}_1((1-\gamma) \sum_{t=0}^{\infty} \gamma^t \hat{\mu}_t^{N_\mu}, \mu)] + \mathbb{E}[\mathcal{W}_1((1-\gamma) \sum_{t=0}^{\infty} \gamma^t \hat{\nu}_t^{N_\nu}, \nu)]
\end{aligned}
\tag{49}
$$

For simplicity, we denote $\hat{\mu}_\infty = (1-\gamma) \sum_{t=0}^{\infty} \gamma^t \hat{\mu}_t^{N_\mu}$ (similarly $\hat{\nu}_\infty$) and $\hat{\mu}_T = \rho \sum_{t=0}^{T} \gamma^t \hat{\mu}_t^{N_\mu}$ (similarly $\hat{\nu}_T$), where $\rho = \frac{1}{\sum_{t=0}^{T} \gamma^t} = \frac{1-\gamma}{1-\gamma^{T+1}}$. Using Theorem 4 in (Gibbs & Su, 2002), the 1-Wasserstein metric $\mathcal{W}_1$ and the total variation distance $d_{TV}$ satisfy the following,

$$
\begin{aligned}
\mathcal{W}_1(\hat{\mu}_\infty, \hat{\mu}_T) &\leq diam(\mathcal{Z}) \cdot d_{TV}(\hat{\mu}_\infty, \hat{\mu}_T) \\
&= diam(\mathcal{Z}) \cdot \frac{1}{2} \sum_{z \in \mathcal{Z}} |\hat{\mu}_\infty(z) - \hat{\mu}_T(z)|
\end{aligned}
\tag{50}
$$

However,

$$
\hat{\mu}_\infty - \hat{\mu}_T = (1-\gamma)\sum_{t=0}^{\infty}\gamma^t\hat{\mu}_t^{N_\mu} - \frac{1-\gamma}{1-\gamma^{T+1}}\sum_{t=0}^{T}\gamma^t\hat{\mu}_t^{N_\mu}
$$

$$
= (1-\gamma)\sum_{t=0}^{\infty}\gamma^t\hat{\mu}_t^{N_\mu} - \frac{1-\gamma}{1-\gamma^{T+1}}\sum_{t=0}^{T}\gamma^t\hat{\mu}_t^{N_\mu}
$$

$$
+ (1-\gamma)\sum_{t=0}^{T}\gamma^t\hat{\mu}_t^{N_\mu} - (1-\gamma)\sum_{t=0}^{T}\gamma^t\hat{\mu}_t^{N_\mu}
$$

$$
= (1-\gamma)\left(\sum_{t=0}^{\infty}\gamma^t\hat{\mu}_t^{N_\mu} - \sum_{t=0}^{T}\gamma^t\hat{\mu}_t^{N_\mu}\right) + \left((1-\gamma) - \frac{1-\gamma}{1-\gamma^{T+1}}\right)\sum_{t=0}^{T}\gamma^t\hat{\mu}_t^{N_\mu} \qquad (51)
$$

$$
= (1-\gamma)\sum_{t=T+1}^{\infty}\gamma^t\hat{\mu}_t^{N_\mu} - \gamma^{T+1}\frac{1-\gamma}{1-\gamma^{T+1}}\sum_{t=0}^{T}\gamma^t\hat{\mu}_t^{N_\mu}
$$

$$
\leq (1-\gamma)\sum_{t=T+1}^{\infty}\gamma^t\hat{\mu}_t^{N_\mu}
$$

$$
= \gamma^{T+1}\frac{1-\gamma}{\gamma^{T+1}}\sum_{t=T+1}^{\infty}\gamma^t\hat{\mu}_t^{N_\mu}
$$

$$
= \gamma^{T+1}\hat{\mu}_{T+1,\infty}
$$

where $\frac{1-\gamma}{\gamma^{T+1}}$ normalizes $\sum_{t=T+1}^{\infty}\gamma^t\hat{\mu}_t^{N_\mu}$. We utilize Equation (51) in Equation (50) as,

$$
\mathcal{W}_1(\hat{\mu}_\infty, \hat{\mu}_T) \leq diam(\mathcal{Z}) \cdot \frac{1}{2}\sum_{z \in \mathcal{Z}}|\hat{\mu}_\infty(z) - \hat{\mu}_T(z)|
$$

$$
\leq diam(\mathcal{Z}) \cdot \frac{1}{2}\sum_{z \in \mathcal{Z}}|\gamma^{T+1}\hat{\mu}_{T+1,\infty}(z)| \qquad (52)
$$

$$
= \frac{\gamma^{T+1}}{2}diam(\mathcal{Z})
$$

Equation (52) also applies for $\mathcal{W}_1(\hat{\nu}_\infty, \hat{\nu}_T)$, therefore by substituting these into Equation (49),

$$
\text{R.H.S} \leq \mathbb{E}[\mathcal{W}_1((1-\gamma)\sum_{t=0}^{\infty}\gamma^t\hat{\mu}_t^{N_\mu}, \mu)] + \mathbb{E}[\mathcal{W}_1((1-\gamma)\sum_{t=0}^{\infty}\gamma^t\hat{\nu}_t^{N_\nu}, \nu)] + \gamma^{T+1}diam(\mathcal{Z})
$$

$$
= \mathbb{E}[\mathcal{W}_1((1-\gamma)\sum_{t=0}^{\infty}\gamma^t\hat{\mu}_t^{N_\mu}, (1-\gamma)\sum_{t=0}^{\infty}\gamma^t\mu_t)]
$$

$$
+ \mathbb{E}[\mathcal{W}_1((1-\gamma)\sum_{t=0}^{\infty}\gamma^t\hat{\nu}_t^{N_\nu}, (1-\gamma)\sum_{t=0}^{\infty}\gamma^t\nu_t)] + \gamma^{T+1}diam(\mathcal{Z}) \qquad (53)
$$

$$
\leq (1-\gamma)\sum_{t=0}^{\infty}\gamma^t\left(\mathbb{E}[\mathcal{W}_1(\hat{\mu}_t^{N_\mu}, \mu_t)] + \mathbb{E}[\mathcal{W}_1(\hat{\nu}_t^{N_\mu}, \nu_t)]\right) + \gamma^{T+1}diam(\mathcal{Z}).
$$

By substituting Equation (45) into Equation (53)

$$
\text{R.H.S} \leq (1-\gamma)\sum_{t=0}^{\infty}\gamma^t\left(\mathcal{E}_2 N_\mu^{-\frac{1}{2}} + \mathcal{E}_2 N_\nu^{-\frac{1}{2}}\right) + \gamma^{T+1}diam(\mathcal{Z})
$$

$$
= \mathcal{E}_2\left(N_\mu^{-\frac{1}{2}} + N_\nu^{-\frac{1}{2}}\right) + \gamma^{T+1}diam(\mathcal{Z}) \qquad (54)
$$

Therefore, Equation (47) becomes:

$$
\mathbb{E}[|\mathcal{W}_1(\hat{\mu}, \hat{\nu}) - \mathcal{W}_1(\mu, \nu)|] \leq \mathcal{E}_2\left(N_\mu^{-\frac{1}{2}} + N_\nu^{-\frac{1}{2}}\right) + \gamma^{T+1}diam(\mathcal{Z}) \qquad (55)
$$

**Over the full trajectory in the occupancy measure space.** The true distance between consecutive policies $\pi_i$ and $\pi_{i+1}$ after an update is $\mathcal{W}_1(v_{\pi_i}, v_{\pi_{i+1}})$, which is induced by the $i^{th}$ policy update. We estimate this distance using datasets of the policies, i.e. approximated distributions, using $\mathcal{W}_1(\hat{v}_{\pi_i}, \hat{v}_{\pi_{i+1}})$.

For $M$ roll out episodes of each $\pi_i$, we use Equation (55), with $N_\mu = N_\nu = M$, to derive the following error bounds,

$$\mathbb{E}\left[\left|\mathcal{W}_1(v_{\pi_i}, v_{\pi_{i+1}}) - \mathcal{W}_1(\hat{v}_{\pi_i}, \hat{v}_{\pi_{i+1}})\right|\right] \leq 2\mathcal{E}_2 M^{-\frac{1}{2}} + \gamma^{T+1} diam(\mathcal{Z}) \tag{56}$$

which is consistent with learning from $\mathcal{D}_{\pi_i}$ and then $\mathcal{D}_{\pi_{i+1}}$. By summing sequentially through policies encountered during RL training, we compute the total distance over a path of $N$ segments obtained via policy updates:

$$\sum_{i=0}^{N-1} \mathbb{E}\left[\left|\mathcal{W}_1(v_{\pi_i}, v_{\pi_{i+1}}) - \mathcal{W}_1(\hat{v}_{\pi_i}, \hat{v}_{\pi_{i+1}})\right|\right] \leq 2N\mathcal{E}_2 M^{-\frac{1}{2}} + N\gamma^{T+1} diam(\mathcal{Z}) \tag{57}$$

Since $|\sum_t x_t| \leq \sum_t |x_t|$ then,

$$\mathbb{E}\left[\left|\sum_{i=0}^{N-1} \mathcal{W}_1(v_{\pi_i}, v_{\pi_{i+1}}) - \sum_{i=0}^{N-1} \mathcal{W}_1(\hat{v}_{\pi_i}, \hat{v}_{\pi_{i+1}})\right|\right] \leq \frac{2N\mathcal{E}_2}{\sqrt{M}} + N\gamma^{T+1} diam(\mathcal{Z}) \tag{58}$$

### A.7.2 FINITE HORIZON MDPS

**Occupancy Measure Estimated Error.** Consider two occupancy measures $\mu = \frac{1}{H}\sum_{t=1}^H \mu_t$ and $\nu = \frac{1}{H}\sum_{t=1}^H \nu_t$ with estimates $\hat{\mu} = \frac{1}{H}\sum_{t=1}^H \hat{\mu}_t^{N_\mu}$ and $\hat{\nu} = \frac{1}{H}\sum_{t=1}^H \hat{\nu}_t^{N_\nu}$. From Equation (46), we have

$$\begin{aligned}
\mathbb{E}[&|\mathcal{W}_1(\hat{\mu}, \hat{\nu}) - \mathcal{W}_1(\mu, \nu)|] \\
&\leq \mathbb{E}[\mathcal{W}_1(\hat{\mu}, \mu) + \mathcal{W}_1(\hat{\nu}, \nu)] \\
&= \mathbb{E}[\mathcal{W}_1(\frac{1}{H}\sum_{t=1}^H \hat{\mu}_t^{N_\mu}, \frac{1}{H}\sum_{t=1}^H \mu_t) + \mathcal{W}_1(\frac{1}{H}\sum_{t=1}^H \hat{\nu}_t^{N_\nu}, \frac{1}{H}\sum_{t=1}^H \nu_t)] \\
&\leq \frac{1}{H}\sum_{t=1}^H \mathbb{E}[\mathcal{W}_1(\hat{\mu}_t^{N_\mu}, \mu_t)] + \frac{1}{H}\sum_{t=1}^H \mathbb{E}[\mathcal{W}_1(\hat{\nu}_t^{N_\nu}, \nu_t)] \\
&\leq \mathcal{E}_2\left(N_\mu^{-\frac{1}{2}} + N_\nu^{-\frac{1}{2}}\right)
\end{aligned} \tag{59}$$

Therefore **for the total path in the occupancy measure space** with $M$ roll out episodes of each $\pi_i$, the error bound is

$$\mathbb{E}\left[\left|\sum_{i=0}^{N-1} \mathcal{W}_1(v_{\pi_i}, v_{\pi_{i+1}}) - \sum_{i=0}^{N-1} \mathcal{W}_1(\hat{v}_{\pi_i}, \hat{v}_{\pi_{i+1}})\right|\right] \leq \frac{2N\mathcal{E}_2}{\sqrt{M}} \tag{60}$$

by assigning $N_\mu = N_\nu = M$ in Equation (59), which concludes the proof.

### A.8 PROOF OF PROPOSITION 5

By definition of $\eta_{sub}$, we get

$$\begin{aligned}
\eta_{sub} &= \frac{\sum_{i=0}^{N-2} \mathcal{W}_1(v_{\pi_i}, v_{\pi_{i+1}}) + \mathcal{W}_1(v_{\pi_{N-1}}, v_{\pi_N})}{\mathcal{W}_1(v_{\pi_0}, v_{\pi_N})} \\
&= \frac{\sum_{i=0}^{N-2} \mathcal{W}_1(v_{\pi_i}, v_{\pi_{i+1}}) + \mathcal{W}_1(v_{\pi_{N-1}}, v_{\pi_N})}{\mathcal{W}_1(v_{\pi_0}, v_{\pi^*})} \times \frac{\mathcal{W}_1(v_{\pi_0}, v_{\pi^*})}{\mathcal{W}_1(v_{\pi_0}, v_{\pi_N})} \\
&\geq \frac{\sum_{i=0}^{N-2} \mathcal{W}_1(v_{\pi_i}, v_{\pi_{i+1}}) + \mathcal{W}_1(v_{\pi_{N-1}}, v_{\pi^*}) - \mathcal{W}_1(v_{\pi_N}, v_{\pi^*})}{\mathcal{W}_1(v_{\pi_0}, v_{\pi^*})} \times \frac{\mathcal{W}_1(v_{\pi_0}, v_{\pi^*})}{\mathcal{W}_1(v_{\pi_0}, v_{\pi_N})} \\
&= \left(\eta - \frac{\mathcal{W}_1(v_{\pi_N}, v_{\pi^*})}{\mathcal{W}_1(v_{\pi_0}, v_{\pi_N})}\right) \frac{\mathcal{W}_1(v_{\pi_0}, v_{\pi^*})}{\mathcal{W}_1(v_{\pi_0}, v_{\pi_N})}.
\end{aligned} \tag{61}$$

The inequality above is true due to the triangle inequality $\mathcal{W}_1(v_{\pi_{N-1}}, v_{\pi_N}) + \mathcal{W}_1(v_{\pi_N}, v_{\pi^*}) \geq \mathcal{W}_1(v_{\pi_{N-1}}, v_{\pi^*})$.

By applying triangle inequality, we also get

$$\mathcal{W}_1(v_{\pi_0}, v_{\pi^*}) + \mathcal{W}_1(v_{\pi_N}, v_{\pi^*}) \geq \mathcal{W}_1(v_{\pi_0}, v_{\pi_N}).$$

This implies that

$$\frac{\mathcal{W}_1(v_{\pi_0}, v_{\pi^*})}{\mathcal{W}_1(v_{\pi_0}, v_{\pi_N})} \geq 1 - \frac{\mathcal{W}_1(v_{\pi_N}, v_{\pi^*})}{\mathcal{W}_1(v_{\pi_0}, v_{\pi_N})}. \tag{62}$$

Equation (61) and Equation (62) together yield

$$
\begin{aligned}
\eta_{sub} &\geq \left( \eta - \frac{\mathcal{W}_1(v_{\pi_N}, v_{\pi^*})}{\mathcal{W}_1(v_{\pi_0}, v_{\pi_N})} \right) \left( 1 - \frac{\mathcal{W}_1(v_{\pi_N}, v_{\pi^*})}{\mathcal{W}_1(v_{\pi_0}, v_{\pi_N})} \right) \\
&= \eta - \frac{\mathcal{W}_1(v_{\pi_N}, v_{\pi^*})}{\mathcal{W}_1(v_{\pi_0}, v_{\pi_N})} - \eta \frac{\mathcal{W}_1(v_{\pi_N}, v_{\pi^*})}{\mathcal{W}_1(v_{\pi_0}, v_{\pi_N})} + \left( \frac{\mathcal{W}_1(v_{\pi_N}, v_{\pi^*})}{\mathcal{W}_1(v_{\pi_0}, v_{\pi_N})} \right)^2 \\
&\geq \eta \left( 1 - \frac{\mathcal{W}_1(v_{\pi_N}, v_{\pi^*})}{\mathcal{W}_1(v_{\pi_0}, v_{\pi_N})} \right) - \frac{\mathcal{W}_1(v_{\pi_N}, v_{\pi^*})}{\mathcal{W}_1(v_{\pi_0}, v_{\pi_N})} \\
&\geq \eta \left( 1 - \frac{2\mathcal{W}_1(v_{\pi_N}, v_{\pi^*})}{\mathcal{W}_1(v_{\pi_0}, v_{\pi_N})} \right).
\end{aligned}
$$

The second last inequality is due to non-negativity of $\left( \frac{\mathcal{W}_1(v_{\pi_N}, v_{\pi^*})}{\mathcal{W}_1(v_{\pi_0}, v_{\pi_N})} \right)^2$. The last inequality is due to the fact that $\eta \geq 1$.

Thus, we conclude that

$$\frac{\eta - \eta_{sub}}{\eta} \leq \frac{2\mathcal{W}_1(v_{\pi_N}, v_{\pi^*})}{\mathcal{W}_1(v_{\pi_0}, v_{\pi_N})}.$$

### A.9  WASSERSTEIN SPACES AS GEODESIC SPACES

Given probability measures $\mu, \nu \in \mathcal{P}(\mathcal{X})$ on a metric space $\mathcal{X} \subset \mathbb{R}^B$ with metric $d_{\mathcal{X}}(x, x')$, the Wasserstein distance $\mathcal{W}_p(\mu, \nu)$ is the minimal transport cost for $c(x, x') = (d_{\mathcal{X}}(x, x'))^p$ with $p \geq 1$ (Villani, 2009). The Wasserstein distance $\mathcal{W}_p(\mu, \nu)$ takes a distance on $\mathcal{X}$ and creates out of it a distance on $\mathcal{P}(\mathcal{X})$(Peyré, 2019). Proposition 5.1 in (Santambrogio, 2015) asserts that $\mathcal{W}_p$ is a distance over $\mathcal{P}(\mathcal{X})$.

**Definition A.9** (Wasserstein Space). (Santambrogio, 2015) *Given a Polish space $\mathcal{X}$, for each $p \in [1, \infty)$, the space $\mathcal{P}(\mathcal{X})$ endowed with the distance $\mathcal{W}_p$ is a Wasserstein space $\mathbb{W}_p$ of order $p$.*

Theorem 5.27 in (Santambrogio, 2015) states that if $\mathcal{X}$ is a convex space, then the space $\mathbb{W}_p$ is a geodesic space (length space). Thus, the geodesic (shortest path distance) between $\mu, \nu \in \mathcal{P}(\mathcal{X})$ is given by $\mathcal{W}_p(\mu, \nu)$ (Kolouri et al., 2017). It was mentioned in Appendix A.7.1 that the RL problems we consider consist of the state-action space $\mathcal{Z} = \mathcal{S} \times \mathcal{A} \in \mathbb{R}^B : B \geq 2$ (subsets of the Euclidean space). Given that Euclidean spaces are convex spaces (Boyd & Vandenberghe, 2004), our space of occupancy measures $\mathcal{M}$ is a Wasserstein space $\mathbb{W}_1 = (\mathcal{M}, \mathcal{W}_1)$ and thus a geodesic space. Therefore, $\mathcal{W}_1(\mu, \nu)$ measures the shortest path on the surface of the manifold $\mathcal{M}$ between probability distributions $\mu$ and $\nu$.

# B ADDITIONAL EXPERIMENTAL ANALYSIS AND RESULTS

## B.1 ENVIRONMENT DESCRIPTION

**2D-Gridworld** environment of size 5x5 with actions: {up, right, down, left}. The start and goal states are always located at top-left and bottom-right, respectively. A) *Deterministic, dense rewards setting*: State transitions are deterministic. The reward function is given by $\|s_t - s_g\|_1$, where $s_t$ is the agent state at timestep $t$ and $s_g$ is the goal state. B) *Deterministic, sparse rewards setting*: State transitions are deterministic and all states issue a reward of -0.04 except the goal state with reward of 1. C) *Stochastic, dense rewards setting*: Actions have a probability of 0.8 in the instructed direction and 0.1 in each adjacent direction. Reward function is as defined in setting A.

**2D-Gridworlds (Task Difficulty).** Figure 6 depicts the configurations of the 5 tasks that were used to assess ESL with respect to task hardness. They are all deterministic with actions: {up, right, down, left}, and mostly have the start-state at the top-left and the goal-state at the bottom-right - with only one task that has the goal-state at the center. In the order of appearance: a) *[5x5] dense*: has size 5x5 and dense rewards, b) *[5x5] sparse (hard)*: has size 5x5 and sparse rewards, c) *[5x5] sparse (easy)*: has size 5x5, sparse rewards, and goal-state at the center, d) *[15x15] dense*: has size 15x15 and dense rewards, and e) *[15x15] sparse*: has size 15x15 and sparse rewards. The reward functions for both dense and sparse rewards are as previously described above for **2D-Gridworld**.

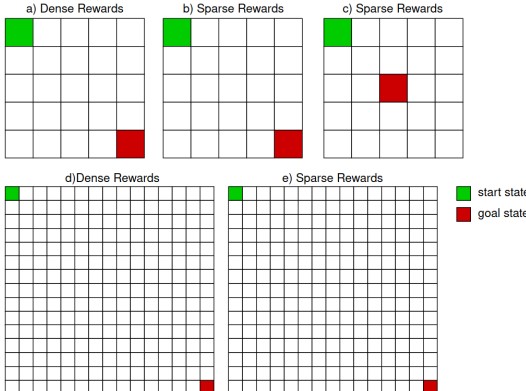

Figure 6: Five gridworld tasks with the same action space, but different rewards, state space and location of the goal state.

## B.2 OMR(K): OMR OVER NUMBER OF UPDATES

OMR is defined for the entire policy trajectory by Equation 6 as,

$$\kappa \triangleq \frac{\sum_{k \in K^+} \mathcal{W}_1(v_{\pi_k}, v_{\pi_{k+1}})}{\sum_{k=0}^{N-1} \mathcal{W}_1(v_{\pi_k}, v_{\pi_{k+1}})}.$$

To observe how it changes with respect to updates, we compute OMR from update $i$ onwards till the end of the learning trajectory, i.e. over subsequences with a decreasing number of policy updates with increasing $i$, using:

$$\kappa(i) \triangleq \frac{\sum_{k \in K^+, k \geq i} \mathcal{W}_1(v_{\pi_k}, v_{\pi_{k+1}})}{\sum_{k=i}^{N-1} \mathcal{W}_1(v_{\pi_k}, v_{\pi_{k+1}})}, \text{ such that } i \in [0, N - T] \tag{63}$$

where $T \approx 0.9N$ to ensure that the last subsequence of policy updates have at least 10% of the total updates in the trajectory.

### B.3 COMPUTATION OF OCCUPANCY MEASURES

The finite-horizon occupancy measure is defined as (Altman, 1999),

$$v_\pi^H(s, a) = \frac{1}{H} \sum_{t=1}^{H} \mathbb{P}(s_t = s, a_t = a \mid \pi, \mu)$$

for which the probability of the state-action pair selected is time-dependent. If we restrict our analysis to stationary policies where $\pi(a_t|s_t) = \pi(a|s)$, then the probability of the state-action pair becomes time-independent and thus

$$v_\pi^H(s, a) = \mathbb{P}(s, a \mid \pi, \mu)$$

This implies that the use of stationary policies in finite-horizon MDPs, as observed in practice with many episodic MDPs (Memmel et al., 2022; Aleksandrowicz & Jaworek-Korjakowska, 2023; Liu, 2023), induces stationary occupancy measures - where the expected number of state-action pair visits are independent of the time-step. Work by (Bojun, 2020) provides extensive details about the existence of stationarity in episodic MDPs and shows (in Theorem 4) that

$$\mathbb{E}_{(s,a) \sim v_\pi^H} \left[ \bar{\mathcal{R}}(s, a) \right] = \frac{\mathbb{E}_{\zeta \sim M_\pi} \left[ \sum_{t=1}^{H(\zeta)} R(s_t, a_t) \right]}{\mathbb{E}_{\zeta \sim M_\pi} \left[ H(\zeta) \right]} \tag{64}$$

where $\zeta$ is the episodic state-action pair trajectory, $H(\zeta)$ is the episode length corresponding to $\zeta$, and $M_\pi$ is the markov chain induced by policy $\pi$. We verified the correctness of our $v_\pi^H$ computation by calculating the relative error derived from Equation 65 to check its validity. The relative error is given as

$$\text{Rel. Error \%} = 100 * \frac{\mathbb{E}_{(s,a) \sim v_\pi^H} \left[ \bar{\mathcal{R}}(s, a) \right] \mathbb{E}_{\zeta \sim M_\pi} \left[ H(\zeta) \right] - \mathbb{E}_{\zeta \sim M_\pi} \left[ \sum_{t=1}^{H(\zeta)} R(s_t, a_t) \right]}{\mathbb{E}_{\zeta \sim M_\pi} \left[ \sum_{t=1}^{H(\zeta)} R(s_t, a_t) \right]} \tag{65}$$

Figure 7 depicts Rel. Error% vs number of updates in the stochastic 2D-Gridworld environment with dense rewards. We observe that increasing the number of rollouts $M$ reduces the estimation error of $v_\pi^H$. For $M = 10$, the absolute relative error can be as high as 50% with the mean less than 10%. While for $M = 500$, the maximum absolute relative error is 4%.

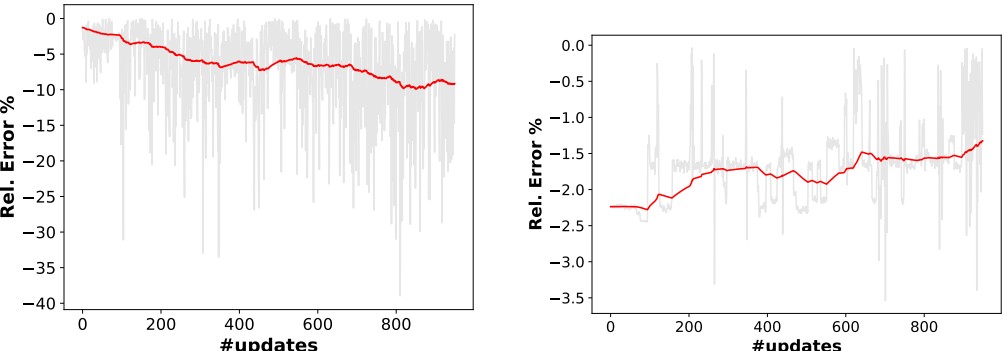

Figure 7: Rel. Error% vs number of updates plots in the 2D-Gridworld environment where $v_\pi^H$ is estimated using $M = 10$ rollouts (left) and $M = 500$ rollouts (right).

### B.4 EFFECTS OF THE NUMBER OF ROLLOUTS - SAC

The policy dataset $D_{\pi_i}$ in a deterministic environment is made up of (s,a) pairs generated from a single episode of the policy $\pi_i$. In a deterministic environment, this sequence remains the same across repeats of episodes, for each policy $\pi_i$ (deterministic) at update step $i$. Therefore, a single

rollout is sufficient to estimate the occupancy measure $v_{\pi_i}$. In a stochastic environment, rollouts are impacted by the environment's stochasticity. Thus, multiple rollouts are needed to estimate the occupancy measure accurately. As the number of rollouts increases, the occupancy measure should converge and become less noisy.

Table 4 shows that, in a stochastic setting, the ESL values converge as the number of rollouts increases. OMR appears to be invariant across various the number of rollouts, and the mean number of updates appear to be consistent around 2900 (with exception for #rollouts = 1). The results indicate that from about 6 rollouts, the estimated occupancy measures become less noisy. This aligns with Equation (58), which shows that increasing the number of rollouts reduces estimation error.

| #rollouts | ESL | OMR | UC |
|---|---|---|---|
| 1 | 849.1±468.5 | 0.500±0.004 | 1849±742.2 |
| 3 | 618.6±257.3 | 0.501±0.005 | 2413±1397 |
| 6 | 445.4±245.8 | 0.501±0.042 | 2462±2043 |
| 9 | 428.1±234.4 | 0.503±0.004 | 2281±1743 |

Table 4: Evaluation of SAC algorithm in the **stochastic, dense-rewards setting** for 5x5 gridworld with **40 maximum steps per episode** across various number of rollouts. The effects of #rollouts on the Effort of Sequential Learning (ESL), Optimal Movement Ratio (OMR), and number of updates to convergence (UC) are observed.

### B.5 $\eta$ VS $\eta_{sub}$

We compare ESL when the optimal policy was reached, denoted $\eta$, versus when it was not, denoted $\eta_{sub}$, in Tables 5 and 6. First, we observe that the number of rollouts impacts the metric values. Second, $\eta_{sub}$ values are always greater than $\eta$ values. Note that UCRL2 and PSRL update their policies only at the end of each episode, whereas SAC and DQN update theirs after each time step. Hence, $UC_{sub} = 499$ for both UCRL2 and PSRL.

The ESL values (both $\eta$ and $\eta_{sub}$) in Table 6 are lower than those in Table 5, as expected since more data samples reduce estimation error. The distance from the initial policies to the final polices are not so different. Using both Tables 5 and 6, we notice that comparing algorithms with $\eta_{sub}$ yields the same efficiency ranking (e.g. PSRL, UCRL, SAC and DQN) as $\eta$. This indicates that $\eta_{sub}$ reliably predicts results provided by $\eta$ for comparing algorithms.

The results presented in Table 2 for stochastic dense-rewards setting are consistent with those in Table 6 because the number of rollouts used was Nr = 6.

| Algo. | $\eta$ | $\eta_{sub}$ | d | c | UC | $UC_{sub}$ |
|---|---|---|---|---|---|---|
| SAC | 849± 468 | 3623± 4166 | 5.63± 1.50 | 5.26± 2.10 | 1850± 742 | 7451± 3535 |
| UCRL2 | 230± 155 | 613± 999 | 5.65± 0.93 | 5.45± 2.15 | 284± 180 | 499± 0.0 |
| PSRL | 86.2± 44.4 | 389± 102 | 4.96± 1.26 | 5.29± 1.49 | 97.2± 52.5 | 499± 0.0 |
| DQN | 564± 478 | 3911± 1710 | 5.52± 1.39 | 6.54± 2.05 | 1213± 1061 | 9097± 1904 |

Table 5: Evaluation of algorithms in the **stochastic, dense-rewards setting** for 5x5 gridworld with **40 maximum steps per episode** with the number of rollouts Nr = 1. The total number of training episodes is 500. When the algorithm converged at optimality, $\eta$ is the *Effort of Sequential Learning*, $d = \mathcal{W}_1(\pi_0, \pi^*)$ is distance from initial policy to the optimal policy, and UC is the number of updates to convergence. When the algorithm did not converge at the optimal policy, rather a non-optimal $\pi_N$, we use $\eta_{sub}$, $c = \mathcal{W}_1(\pi_0, \pi_N)$, and $UC_{sub}$ to denote the aforementioned quantities. 40 training trials were used.

| Algo. | $\eta$ | $\eta_{sub}$ | d | c | UC | $UC_{sub}$ |
|---|---|---|---|---|---|---|
| SAC | 445± 246 | 853± 127 | 5.63± 1.23 | 7.26± 1.45 | 2463± 2043 | 6293± 441 |
| UCRL2 | 198± 121 | 510± 274 | 5.36± 0.84 | 4.58± 1.90 | 268± 155 | 499± 0.0 |
| PSRL | 55.4± 33.6 | 361± 43.6 | 4.97± 1.34 | 3.91± 0.48 | 76.1± 50.6 | 499± 0.0 |
| DQN | 458± 311 | 1971± 250 | 4.88± 1.06 | 6.52± 0.31 | 1586± 1077 | 13713± 6907 |

Table 6: Evaluation of algorithms in the **stochastic, dense-rewards setting** for 5x5 gridworld with **40 maximum steps per episode** with the number of rollouts Nr = 6. The total number of training episode is 500. When the algorithm converged at optimality, $\eta$ is the *Effort of Sequential Learning*, $d = \mathcal{W}_1(\pi_0, \pi^*)$ is distance from initial policy to the optimal policy, and UC is the number of updates to convergence. When the algorithm did not converge at the optimal policy however some $\pi_N$, we use $\eta_{sub}$, $c = \mathcal{W}_1(\pi_0, \pi_N)$, and $UC_{sub}$ to denote the aforementioned quantities. 40 training trials were used.

### B.6 EFFECTS OF HYPERPARAMETERS - UCRL2

Table 7 illustrates the effects of hyperparameter values in the UCRL2 algorithm. The environment is deterministic dense-rewards setting with 200 training episodes. We observe that high exploration rates ($\delta \to 0$) appear to align with high ESL and UC, while high exploitation rates ($\delta \to 1$) appear to align with low ESL and UC. OMR appears to be invariant across various $\delta$ values.

| $\delta$ | ESL | OMR | UC | SR% |
|---|---|---|---|---|
| 0.1 | 47.76±7.768 | 0.512±0.033 | 62.26±9.977 | 100 |
| 0.3 | 39.29±5.860 | 0.515±0.034 | 58.08±7.746 | 100 |
| 0.5 | 38.26±6.747 | 0.511±0.036 | 56.92±9.111 | 100 |
| 0.7 | 37.48±5.094 | 0.507±0.029 | 56.68±7.460 | 100 |
| 0.9 | 36.40±5.301 | 0.510±0.036 | 54.86±7.326 | 100 |

Table 7: Evaluation of UCRL2 algorithm in the **deterministic, dense-rewards setting** for 5x5 gridworld with **15 maximum steps per episode**. Different confidence parameter $\delta \in (0, 1)$ were evaluated to see their effects on Effort of Sequential Learning (ESL), Optimal Movement Ratio (OMR), number of updates to convergence (UC), and success rate (SR). Note that as $\delta \to 0$, the agent approaches absolute exploration, and with $\delta \to 1$ absolute exploitation.

### B.7 EXTENDED DISCUSSION OF USEFULNESS OF ESL AND OMR

The quantities like regret and number of updates (UC) are outcomes of the exploratory processes, and thus reflect only a partial view of the underlying exploration mechanisms. We propose ESL and OMR to complement regret and number of updates as metrics but not to replace them.

**1. Complementarity of ESL and OMR with respect to UC:**

a. **Case 1.** Let us consider two RL algorithms that reach optimality with the same number of updates, i.e. they have the same UC. *How would one be able to distinguish the exploratory processes of these algorithms?* ESL and OMR are the summary metrics of the policy trajectory during learning. These can reveal which algorithm's exploratory process is more direct versus meandering, smooth versus noisy, or has large versus small coverage area in the policy space (Figures 3 and 4, top rows). Therefore, ESL and OMR quantify with granularity the characteristics of the exploratory process of an RL algorithm for any given environment.

b. **Case 2.** Let us consider the case when optimality is not reached but the maximum number of updates is attained by two RL algorithms. *How would one be able to evaluate the exploratory processes of these algorithms and systematically uncover which exploratory process demonstrate desired characteristics?* Looking into the training trajectories of RL algorithms in an environment and corresponding higher/lower ESLs ($\eta_{sub}$, Section 4.2), we can make a knowledgeable choice of an RL algorithm exhibiting desired characteristics (e.g. high coverage, smooth exploration).

We have shown in Section 4.2 and results in Appendix B.5 that ranking based on suboptimal ESL is aligned with true ESL, and additionally, the visualization of the training trajectories (Figures 3 and 4) can indicate the characteristics of corresponding RL algorithms even when optimal policy is not reached.

c. **Experimental Evidence.** UCRL2 is known to be provably regret-optimal and is designed to continuously explore. SAC does not have such rigorous theoretical guarantees but is known to be practically efficient. In Table 1, by UC, we observe that SAC is significantly suboptimal than UCRL2. But SAC has lower ESL than UCRL2 as its exploration is smoother. Additionally, OMR for SAC is higher than that of UCRL2. They together indicate that SAC takes smoother but larger number of policy transitions aligned to optimal direction for exploration, while UCRL2 exhibits bigger policy changes and in diverse manner trying to cover the environment faster.

## 2. Complementarity of ESL and OMR with respect to Regret:

UCRL2 and PSRL have the same order of regret bound (Osband et al., 2013). But PSRL leads to smoother policy transitions that are much more orientated towards optimality (as shown in Figure 3), while UCRL2 leads to less smooth policy transitions that do not taper as it approaches optimality. This information is not evident from regret but from corresponding ESLs and OMRs (Table 1).

## 3. Insights for Algorithm Design:

Knowing ESL (or suboptimal ESL) and OMR can assist with developing algorithms that emphasize certain exploratory characteristics. We can develop algorithms with grades of coverage or directness, while also being able to visualize this. Ultimately, depending on the environment, we can choose which characteristics of exploratory process are well suited. In contrast, looking only at the final outcomes of RL algorithms like regret and number of updates does not include these nuances.

## C    Specifications of the RL Algorithms under Study

### C.1    Methods for simulation results (Discrete MDP)

**Model parameter initialisation.** We initialised model parameters for deep learning RL algorithms like DQN and SAC by uniformly sampling weight values between $-3 \cdot 10^{-4}$ and $3 \cdot 10^{-4}$ and the biases at $0$. For tabular Q-learning algorithms, we randomly initialized the Q-values between $-1.0$ and $1.0$. For UCRL and PSRL, the policy model was randomly initialized. Note that all Wasserstein distances were computed using a python package POT (Flamary et al., 2021). Additionally, L1 norm was used in our Wasserstein metric cost function as the ground metric for the 2D gridworld environment.

**Results in Figure 3.** The problem setting was deterministic with dense-rewards and 15 maximum number of steps per episode. The total number of episodes was 200. The convergence criterion was satisfied when maximum returns were produced by an algorithm over 5 consecutive updates. The results showcase a single representative run of each algorithm. The confidence parameter $\delta = 0.1$ was utilized for UCRL2. The $\alpha$ parameter for SAC was autotuned using the approach in (Haarnoja et al., 2019) along with hyperparameters described in Table 8. While DQN began with $\epsilon = 1.0$ and the value decayed as $\epsilon[t + 1] = \max\{0.9999 \times \epsilon[t], 0.0001\}$. Table 9 shows hyperparameters for DQN. Note that the ADAM (Kingma & Ba, 2017) optimizer was used in all the neural network models.

Table 8: SAC Hyperparameters.

| Parameter | Value |
|---|---|
| learning rate | $5 \cdot 10^{-4}$ |
| discount($\gamma$) | 0.99 |
| replay buffer size | $10^4$ |
| number of hidden layers (all networks) | 1 |
| number of hidden units per layer | 32 |
| number of samples per minibatch | 64 |
| nonlinearity | ReLU |
| entropy target | -4 |
| target smoothing coefficient ($\tau$) | 0.01 |
| target update interval | 1 |
| gradient steps | 1 |
| initial exploration steps before model starts updating | 500 |

Table 9: DQN Hyperparameters.

| Parameter | Value |
|---|---|
| learning rate | $5 \cdot 10^{-2}$ |
| discount($\gamma$) | 0.99 |
| replay buffer size | $10^4$ |
| number of hidden layers (all networks) | 1 |
| number of hidden units per layer | 32 |
| number of samples per minibatch | 64 |
| nonlinearity | ReLU |
| target smoothing coefficient ($\tau$) | 0.001 |
| target update interval | 1 |
| gradient steps | 1 |
| initial exploration steps before $\epsilon$ decays | 500 |

**Results in Tables 1 and 2.** The problem settings had 40 maximum number of steps per episode, and the convergence criterion was satisfied when maximum returns were produced by an algorithm over 5 consecutive updates. The means and standard deviations for each algorithm were computed over 50 runs. The total number of episodes was 200 for results in Table 1, and 500 in Table 2. For results

in Figure 5, the Q-learning with decaying $\epsilon$-greedy where $\epsilon = 0.9$ was employed in the gridworld tasks described in Appendix B.1. A convergence criterion of 50 consecutive model updates with maximum returns was utilized. We aggregated the result over 40 training trials and the maximum number of steps per episode was 60.

### C.2 METHODS FOR SIMULATION RESULTS (CONTINUOUS MDP)

**Model parameter initialisation.** We initialised model parameters for the deep learning SAC algorithm by uniformly sampling weight values between $-3 \cdot 10^{-4}$ and $3 \cdot 10^{-4}$ and the biases at 0. For the DDPG algorithm, the output layer weight values were initialised using Xavier Initialization (Glorot & Bengio, 2010), while the rest were uniformly sampled between $-3 \cdot 10^{-3}$ and $3 \cdot 10^{-3}$. This was done on both the actor and critic networks. The ADAM (Kingma & Ba, 2017) optimizer was used in all the neural network models. In both algorithms, 1) a discount factor $\gamma = 0.99$ was used, 2) 500 initial steps were taken before updating model weights, and 3) replay buffer size was $10^6$. Tables 10 and 11 display hyperparameters for DDPG and SAC, respectively.

**Results in Figure 4.** The problem setting was Mountain Car continuous (Moore, 1990) with 999 maximum number of steps per episode (Brockman et al., 2016). The total number of training episodes was 100. The convergence criterion was satisfied when maximum returns were produced by an algorithm over 10 consecutive updates. The results showcase a single representative run of each algorithm. For results in **Table 3**, the mean and standard deviations for each algorithm were computed over 5 runs. While RL training was conducted in a continuous state-action space, we discretized it for Wasserstein distance calculations between occupancy measures, using 4 bins for actions and 10 bins for states. Note that all Wasserstein distances were computed using a python package POT (Flamary et al., 2021). Additionally, L2 norm was used in our Wasserstein metric cost function as the ground metric for the Mountain Car environment.

Table 10: DDPG Hyperparameters.

| Parameter | Value |
|---|---|
| number of samples per minibatch | 128 |
| nonlinearity | ReLU |
| target smoothing coefficients ($\tau$) | 0.001 |
| target update interval | 1 |
| gradient steps | 1 |
| number of hidden layers (all networks) | 2 |
| number of hidden units per layer | 64 |
| Actor learning rate | $5 \cdot 10^{-4}$ |
| Critic learning rate | $5 \cdot 10^{-3}$ |

Table 11: SAC Hyperparameters.

| Parameter | Value |
|---|---|
| learning rate | $3 \cdot 10^{-3}$ |
| number of hidden layers (all networks) | 2 |
| number of hidden units per layer | 64 |
| number of samples per minibatch | 128 |
| nonlinearity | ReLU |
| target smoothing coefficient ($\tau$) | 0.001 |
| target update interval | 1 |
| gradient steps | 1 |

# D    SUPPLEMENTARY RESULTS

In this section we present enlarged versions of results in Figure 3 (see Section D.1) and additional plots that support the results in the main paper (see Section D.2). Note that the Github repository of the project is available at [link on acceptance].

## D.1    ENLARGED VISUALISATION OF THE OCCUPANCY MEASURE TRAJECTORIES

Figures 8 - 10 are enlarged versions of enlarged versions of Figure 3. For each algorithm, there is a visualisation of the policy trajectory and visualisation of the state visitation below it.

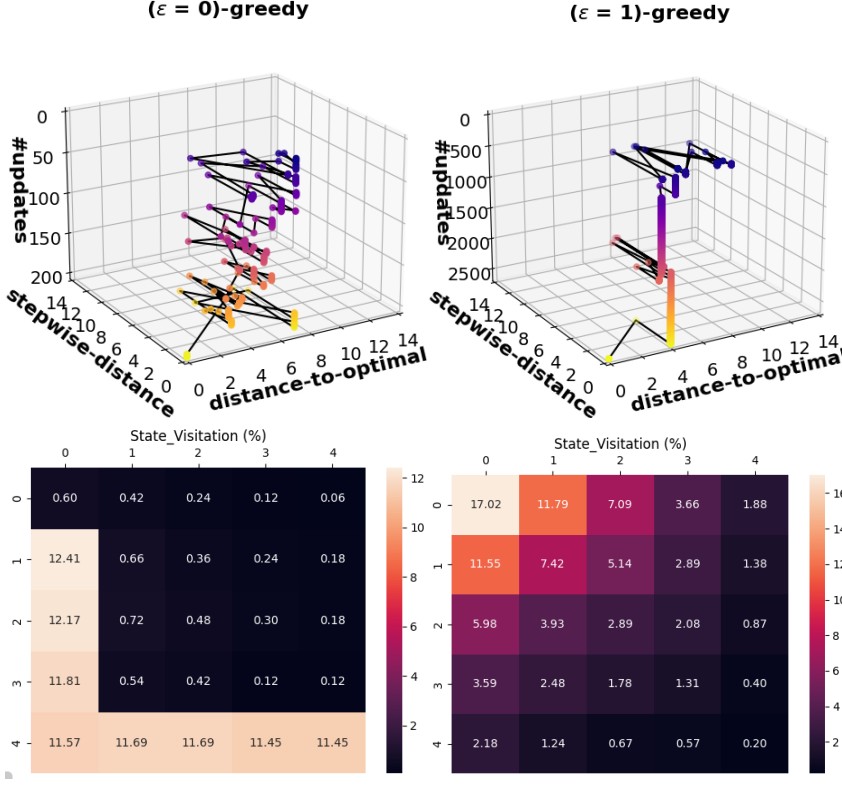

Figure 8: Top row: Scatter plots of *distance-to-optimal* and *stepwise-distance* over updates for $\epsilon(=0)$-greedy and $\epsilon(=1)$-greedy Q-learning. Bottom row: State visitations.

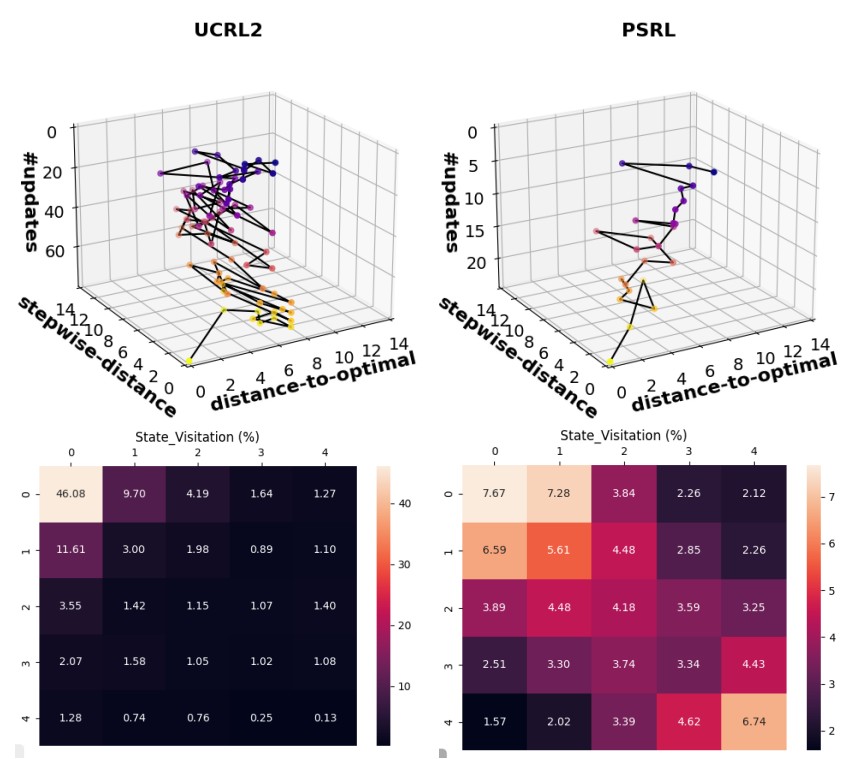

Figure 9: Top row: Scatter plots of *distance-to-optimal* and *stepwise-distance* over updates for UCRL2 and PSRL. Bottom row: State visitations.

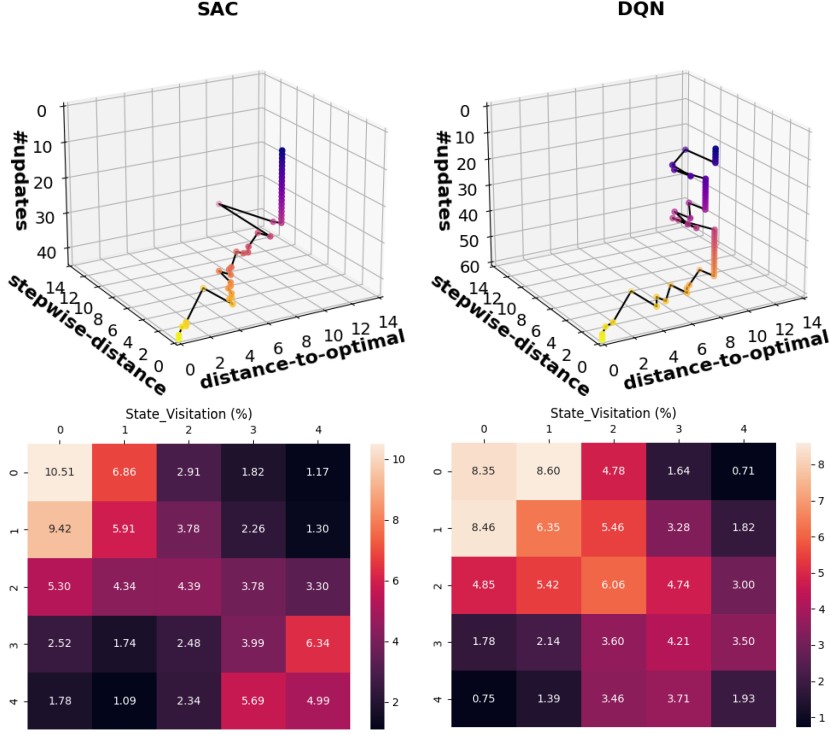

Figure 10: Top row: Scatter plots of *distance-to-optimal* and *stepwise-distance* over updates for SAC and DQN. Bottom row: State visitations.

## D.2 EVOLUTION OF *stepwise-distance*, *distance-to-optimal*, AND OMR($k$)

In this section we present 2 dimensional versions of the policy trajectories in Figures 3 and 4, along with corresponding OMR evolution plots. These are *stepwise-distance* vs. updates, *distance-to-optimal* vs. updates, and OMR($k$) plots for the algorithms. Figure 11 presents plots for the continuous environment Mountain Car, while Figure 12) presents plots for the discrete environment 2D Gridworld.

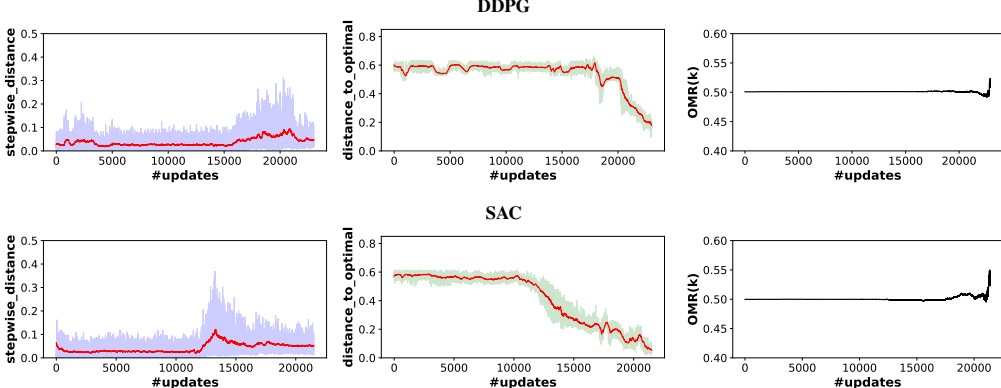

Figure 11: Plots in the first column are *stepwise-distance* vs. number of updates, second column *distance-to-optimal* vs. number of updates, and third OMR($k$) vs. number of updates. Top row plots belong to DDPG algorithm, while bottom row plots belong to SAC.

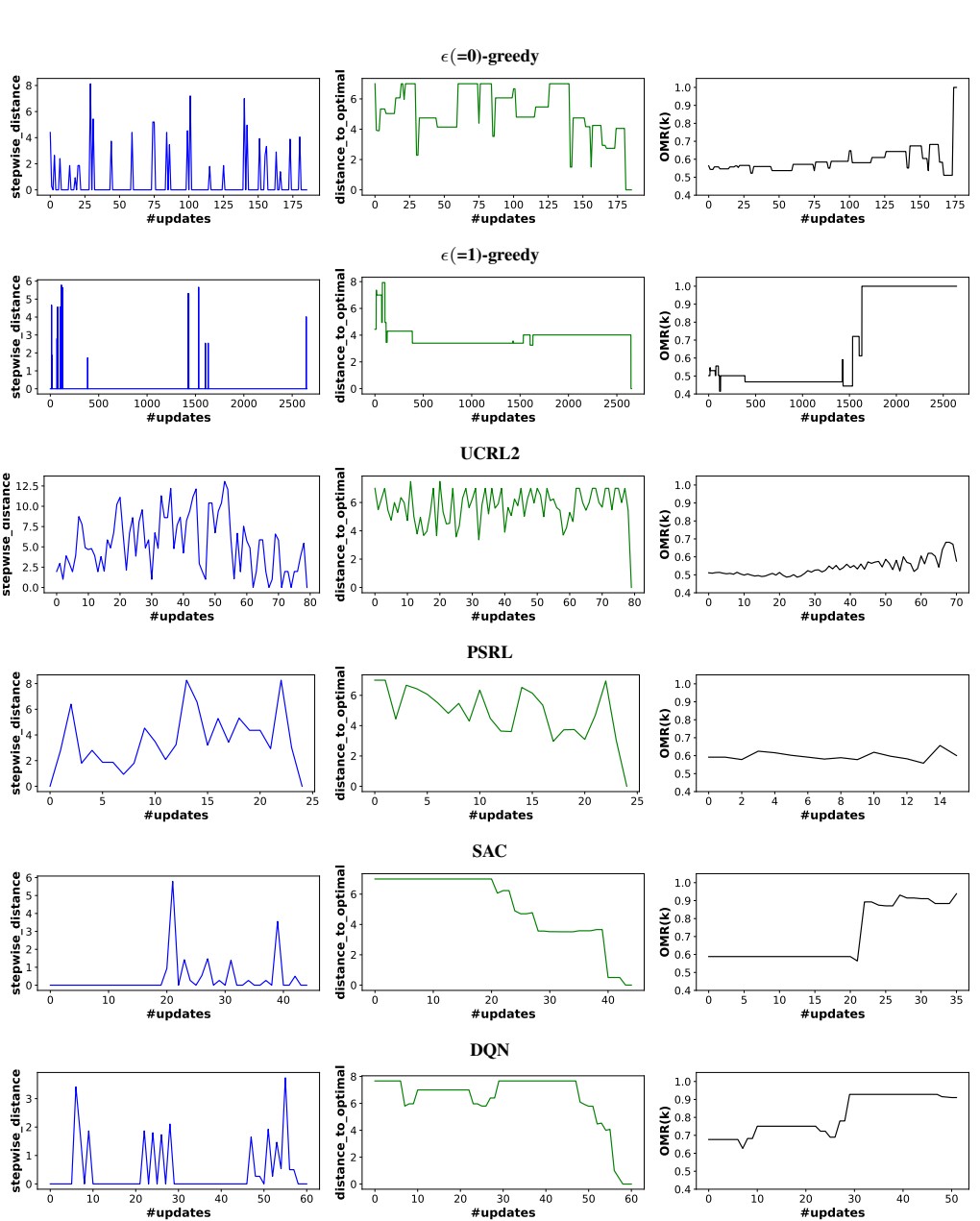

Figure 12: Plots in the first column are *stepwise-distance* vs. number of updates, second column *distance-to-optimal* vs. number of updates, and third OMR($k$) vs. number of updates. The plots in the row belong to algorithms in the following order from top to bottom: $\epsilon(=0)$-greedy, $\epsilon(=1)$-greedy, UCRL2, PSRL, SAC, and DQN.

