# OpenReview forum: "How does Your RL Agent Explore? An Optimal Transport Analysis of Occupancy Measure Trajectories"
_ICLR.cc/2025/Conference — Submitted to ICLR 2025_

### Official Review · Reviewer_rCFe · 2024-10-24

**Soundness:** 2
**Presentation:** 3
**Contribution:** 2
**Rating:** 6
**Confidence:** 2

**Summary:**

This paper proposes two ratios to characterize the learning pattern of RL algorithms, the ESL and OMR. ESL characterize how straight is the training trajectories, which takes smaller values if the occupancy measures admits some straight lines. OMR characterizes the proportion of trajectories which has positive effects toward the optimal policy.

This paper propose some finite-sample analysis in approximating the ESL and OMR with iid samples. They also provide numerical experiments to calculate ESL and OMR values for some popular algorithms in RL literatures.

**Strengths:**

1. This paper is well-written. The proofs are correct.

2. This paper proposes the ESL and OMR notions, which makes contribution in understanding the effectiveness of training dynamics of RL algorithms.

**Weaknesses:**

1. The definition of OMR seems very weird. This notion is not continuous to the training trajectories, which means that the OMR value can change a lot if the model is only perturbed a little (maybe due to randomness or small approximation error). It will be good if there is a more robust notion which serves the same purpose of OMR.

2. According to the finite sample proposition (Proposition 4), the number of samples required to get a good approximation is proportional to the number of states and number of actions, which could be exponentially large for real tasks.

3. This two notions of complexity only characterize the training dynamics, but there still exist some algorithms which enjoys good performances even though the dynamics are volatile. These two notions cannot capture such properties of the algorithms.

**Questions:**

1. Normally the Wasserstein distances are not very easy to calculate, hence people usually use approximation algorithms, e.g. Sinkhorn algorithms to approximate the W-distances. How does this approximation error affects the value of ESL and OMR?

2. Is there any relation between the performance of the algorithm and these two notions of complexity, either theoretically or practically?

---

> ### Author Response · Authors · 2024-11-18
>
> We thank the reviewer for the time and effort spent reviewing. We try to address the concerns and questions here.
>
> **1. Intuitions behind OMR.** 1) We agree that OMR is not continuous over the training trajectory as the steps towards optimal changes are stochastic for any randomized RL algorithm. Similar discontinuity emerges when we try to measure and bound the number of pulls of a specific suboptimal action in regret minimization (e.g. for bandits we refer to Sec 4.5. in [1]), but still this quantity helps to measure the suboptimality of an RL algorithm in terms of regret. 2) We understand that the OMR can change due to smaller changes in the training and the environment. But in expectation, it should depend on the RL algorithm's dynamics for a given environment. Thus, we present an average OMR ($\\pm$ standard deviation) over multiple runs of an RL algorithm in a given environment.
>
> **2. Dependence on size of state-action space.** The problem-dependent constant in Proposition 4, i.e. $\mathcal{E}_{2}$, is upper-bounded by the product of the state-action space diameter and a sublinear function of its cardinality (in and following Eq (45), Appendix A.7.1). So, number of samples would not grow exponentially with the cardinality of the state-action space ($|\mathcal{Z}|$). Specifically, in Appendix A.7.1., we state that
>
> 1) $\mathcal{E}_{2} \leq 4B^{1/2} \text{diam}(\mathcal{Z}) \cdot ( 2 + \frac{1}{2} \text{log} |\mathcal{Z}| )$ $\text{, if } B = 2$
> 2) $\mathcal{E}_{2} \leq 4B^{1/2} \text{diam}(\mathcal{Z}) \cdot ( |\mathcal{Z}|^{1/2 - 1/B} \[ 2 + \frac{1}{2^{B/2 -1} - 1} \] )$ $\text{, if } B > 2$
>
> where $|\mathcal{Z}|$ and $diam(\mathcal{Z})$ denote the cardinality and diameter of space $\mathcal{Z}$, respectively, given that $\mathcal{Z} \in \mathbb{R}^{B}$.
>
> **3. Computing ESL and OMR during training.** Exploration and learning of RL algorithms mainly occur during training. Hence, our metrics are defined within this phase. OMR does use the change in distance to optimal (reduction or increase) at each training step. Also, ESL is the ratio of the learning path to the geodesic path from initial to the final optimal policy. Thus both performance and training dynamics are captured by our metrics.
>
> We are not clear about what you mean by "enjoys good performances even though the dynamics are volatile". If you can explain, we would be happy to respond further.
>
> **4. Computing Wasserstein distance.** Thanks for the suggestion. For the environments that we studied, we did not need to use Sinkhorn approximations. This is interesting for future work for environments where Wasserstein is costlier to compute. And if we use Sinkhorn, specifically Online Sinkhorn [2], the approximation error will also be of similar order $\( 1/\sqrt{\\#\text{samples}} \)$ as in Proposition 4.
>
> **5. Exploratory process and performance.** Performance is an outcome of the exploratory process and it frequently occurs (depending on the environment) that different exploratory processes (algorithms) have the same performance. However, ESL and OMR directly capture and evaluate the inherent characteristics of the exploratory processes. For example, regret is a common measure of performance, whereas our measures are complementary. Please see our general comment ("Utility of proposed metrics"). Additionally, Proposition 2 connects distance-to-optimal with regret. Depending on the environment, some characteristics might benefit or harm the performance of the algorithm. It should be noted that these characteristics are also impacted by the nature of the environment (sparse vs. dense, stochastic vs. deterministic) as shown in the experiments.
>
> [1] T. Lattimore and C. Szepesvári. Bandit algorithms. Cambridge University Press, 2020. ISBN 978-1-108-48682-8.
>
> [2] A. Mensch and G. Peyré. Online sinkhorn: Optimal transport distances from sample streams. Advances in Neural Information Processing Systems, 33, 1657-1667, 2020.

---

> > ### Comment · Reviewer_rCFe · 2024-11-24
> >
> > Thanks very much for the authors' response. I have increased the score accordingly.

---

### Official Review · Reviewer_yMaN · 2024-11-03

**Soundness:** 2
**Presentation:** 3
**Contribution:** 2
**Rating:** 3
**Confidence:** 4

**Summary:**

The paper proposes new metrics, effort of sequential learning (ESL) and optimal movement ratio (OMR), that are used to study the sequence of policies, $\pi_{\theta_1}, \ldots, \pi_{\theta_n}$, generated by a reinforcement learning (RL) algorithm by examining the corresponding sequence of state-action occupancy measures, $v_{\pi_{\theta_1}}, \ldots, v_{\pi_{\theta_n}}$, induced by those policies. These metrics consider the 1-Wasserstein distances between consecutive occupancy measures $W_1(v_{\pi_{\theta_k}}, v_{\pi_{\theta_{k+1}}})$ over the entire space of state-action occupancy measures, $\Delta(S \times A)$. Theoretical results are provided stating that the set $\mathcal{M} =$ {$v_{\pi_{\theta}} \ | \ \theta \in \Theta$} $\subset \Delta(S \times A)$ is a differentiable manifold and that regret of an RL algorithm is upper-bounded by the sum of Wasserstein distances, and practical approximations of ESL and OMR computed using finite-rollout datasets are proposed and upper bounds on their approximation error are given. Experimental results are provided on a 5x5 gridworld problem and the MountainCar environment that visualize occupancy measure trajectories of a variety of algorithms (Q-learning, UCRL, PSRL, SAC, DQN), compare their ESR, OMR and total number of updates performed until convergence (UC), and illustrate that ESL tends to increase with task difficulty.

**Strengths:**

Overall, the paper's objective of developing metrics for quantitative comparison of the learning processes of RL algorithms that go beyond the standard sample complexity / total number of update steps and regret is well-motivated and likely of interest to the community. In particular, the idea of achieving this by examining the effort expended in shifting probability mass around in occupancy measure space during the learning process is an interesting one and worth further investigation. Finally, the paper is clearly written and straightforward to understand.

**Weaknesses:**

Despite its strengths, the paper has key weaknesses.

First, the ESR and OMR metrics are misguided, since the effort measured by Wasserstein distance in the space of state-action occupancy measures does not necessarily correspond to effort performed by the algorithm. This arises due to Definition 1 not taking the differential structure of the manifold $\mathcal{M}$ into account: specifically, the distance $d_{\mathcal{X}}$ that the paper uses in the definition of the Wasserstein distance is $\ell$1 distance in the space $\Delta( S \times A)$, instead of a corresponding distance over the surface of the manifold $\mathcal{M}$ (see the discussion following eq. (2)). This means that ESL and OMR may misrepresent occupancy measures that are far apart (or close together) over the surface of $\mathcal{M}$ as close together (or far apart) by only considering their distance over $\Delta( S \times A)$.

To see this, suppose that two different algorithms, algorithm 1 and algorithm 2, yield a sequence of policies $\pi_{\theta_1}, \ldots, \pi_{\theta_m}$ and $\pi_{\theta_1}', \ldots, \pi_{\theta_n}'$, respectively, where $m \ll n$ and $\pi_{\theta_m} = \pi_{\theta_n}' = \pi^*$. Suppose furthermore that the occupancy measure trajectories $v_{\pi_{\theta_1}}, \ldots, v_{\pi_{\theta_m}}$ and $v_{\pi_{\theta_1}'}, \ldots, v_{\pi_{\theta_n}'}$ are such that $C = \sum_{k=1}^{m-1} W_1(v_{\pi_{\theta_k}}, v_{\pi_{\theta_{k+1}}}) \gg 1$, while $C' = \sum_{k=1}^{n-1} W_1(v_{\pi_{\theta_k}'}, v_{\pi_{\theta_{k+1}}'}) = 1$ (see eq. (4) for $C$ definition). Without imposing special structure on the policy parametrization, there is nothing preventing such a situation from arising. In this case, we have that the ESL $\eta$ of algorithm 1 (see ESL definition in eq. (5)) satisfies $\eta \gg 1$, while the ESL $\eta'$ of algorithm 2 satisfies $\eta' = 1$, despite the fact that algorithm 1 achieved optimality in far fewer steps ($m \ll n$) than algorithm 2. This casts doubt on the usefulness of ESL and OMR as proxies for the efficiency of the learning process. **Notice that this kind of situation actually arises in the experiments provided in the paper:** in Table 1 and the top half of Table 2 the algorithms with lowest UC have large ESL and OMR values, while the algorithms with smallest ESL and OMR values have large UC values. Since ESL and OMR are thus not positively correlated with sample complexity or UC, this raises the question: why are ESL and OMR useful? what insight into the learning process do they provide that sample complexity / UC or regret do not? The answer to these questions is not clear from the paper.

In addition, the experimental results do not provide clear arguments supporting the usefulness of ESL and OMR. While visualizing occupancy measure trajectories in Sec. 5.1 is an interesting exercise, it is not clear how these results support the main goal of providing "insight into the exploration processes of RL algorithms" stated in the abstract. For the reasons described at the end of the preceding paragraph, the results of Sec. 5.2 raise concerns regarding the usefulness of ESL and OMR for comparing the learning processes of RL algorithms. Finally, though Sec. 5.3 does provide evidence that ESL is positively correlated with task difficulty, it also raises an important question: aren't sample complexity / total number of update steps or even regret useful for measuring task complexity as well? why should ESL be preferred?

**Suggestion:** ESL and OMR may be more rigorously grounded by developing a version of Definition 1 that accounts for the structure of $\mathcal{M}$. This will complicate the theoretical and approximation results, but it may be possible to consider versions of those results when the differential structure of $\mathcal{M}$ is particularly simple, such as with tabular or softmax policies.

**Questions:**

1. What information do ESL and OMR provide about the learning process beyond that provided by sample complexity / UC and regret?
2. Why is ESL better for measuring task complexity than sample complexity / UC or regret?

---

> ### Author Response · Authors · 2024-11-18
>
> We thank the reviewer for the time spent reviewing and acknowledging the clarity of our paper and contributions. We refer to the general comment ("Utility of proposed metrics") for detailed discussion on usefulness ESL and OMR. Here, we respond to the other questions/concerns in detail.
>
> **1. Wasserstein distance and geodesics.** 1) The Wasserstein metric incorporates the geometry of the manifold $\mathcal{M}$ through the coupling distribution $d\pi$ (seen in Eq (2)). Additionally, [1,2] show that 1-Wasserstein and $L_1$ distances between probability measures are related but different (see also Fig 5 https://www.stat.cmu.edu/~larry/=sml/Opt.pdf). 2) $(d_{\mathcal{X}})^p$ is the cost function used to define Wasserstein distance, and $d_{\mathcal{X}}$ is the metric defined over the domain $\mathcal{X}$ of the probability distributions. $d_{\mathcal{X}}$ can be any metric reflecting the structure of optimal transport plans in the space $\mathcal{X}$ [3,4]. Thus, we used L1 metric for discrete gridworld environments and L2 for continuous mountaincar.
>
> [5] states that "An important property of $(P(\Omega),W_{p})$ is that it is a geodesic space and that geodesics are easy to characterize." Chapter 5 in [4] provides details about geodesics in the space ($P(\mathcal{X}), W_{p}$) for $p \ge 1$. $P(\Omega)$ and $P(\mathcal{X})$ are probability distribution spaces, in our work denoted as $\mathcal{M}$, with metric $W_{p}$. Thus, $W_1(\mu,\nu)$ measures the geodesic distance (shortest path on the surface of the manifold) between probability distributions $\mu$ and $\nu$.
>
> We will add a section in the Appendices to clarify these points.
>
> **2. ESL and OMR are not correlated with UC.** We do not expect any correlation between number of updates (UC) and ESL/OMR because any policy update, irrespective of its stepwise-distance $W_{1}(v_{\pi_{\theta_k}},v_{\pi_{\theta_{k+1}}})$, may not be in the direction towards an optimal policy. For example, one algorithm may make shorter transitions but in the correct direction with higher UC (e.g. SAC). While another algorithm may make longer transitions yet meander towards the correct direction with lower UC (e.g. UCRL2). Meandering expends notable effort hence higher ESL though it may reduce UC. This is well illustrated in Fig 3. We refer to our general comment ("Utility of proposed metrics") for a detailed discussion.
>
> **3. Insights from visualizing the exploratory processes.** 1) Visualization of exploratory processes is one of the key contributions of our work, as it helps to visually characterize behaviors of RL algorithms and provide visual saliency. There is utility in this visualization as it can enhance understanding of the exploratory process, build intuitions, and allow validation of models. To our knowledge, visualizing RL exploration in a 3D space reflecting the geometry of high-dimensional learning paths has not appeared in the literature. Hence this contribution should not be overlooked. 2) The results in Section 5.1 are consistent with literature but offer insights beyond performance. For example, [6] highlighted that exploration in PSRL is guided by the variance of sampled policies as opposed to optimism in UCRL2. In Fig 3, *we see that the guiding variance in PSRL reduces after every policy update until optimality is reached, while UCRL2 maintains high variance due to optimism*. Figures in [6] have no way of showing this intuitively.
>
> **4. Experimental evidences for ESL and OMR.** We refer to our general comment ("Utility of proposed metrics") explaining the usefulness of ESL and OMR. Specifically, the aim of Sec 5.2 is to use ESL and OMR to compare exploratory processes in familiar environments while revealing which exploratory characteristics are more suited to the problem (sparse vs. dense reward, deterministic vs. stochastic).
>
> **5. ESL to indicate task complexity.** The purpose of Section 5.3 was to test whether ESL scales with task difficulty, but not to propose it as a measure of task complexity. Since ESL captures the effort expended by the exploratory process, it is expected to increase as task difficulty increases - indeed it does.
>
> [1] Indyk, P. and Thaper, N. Fast image retrieval via embeddings. In 3rd international workshop on statistical and computational theories of vision, volume 2, pp. 5, 2003.
>
> [2] A. Gibbs and F. E. Su. On choosing and bounding probability metrics. International Statistical Review / Revue Internationale de Statistique, 2002.
>
> [3] C. Villani. Optimal Transport Old and New. Springer Berlin, Heidelberg, 2009.
>
> [4] F. Santambrogio. Optimal Transport for Applied Mathematicians. Bikhauser Cham, 2015.
>
> [5] S. Kolouri, et. al. Optimal mass transport: Signal processing and machine learning applications. IEEE Signal Processing Magazine, 2017.
>
> [6] I. Osband, B. V. Roy, and D. Russo. (more) efficient reinforcement learning via posterior sampling. In Advances in Neural Information Processing Systems, 2013.

---

> ### Author Response · Authors · 2024-11-25
>
> Dear Reviewer  yMaN,
>
> We believe that we have fully addressed your concerns with our responses and updated pdf. We hope that you can update your rating if satisfied and / or ask any further questions within the 1 day discussion period remaining.
>
> Thanks.

---

### Official Review · Reviewer_GHMN · 2024-11-07

**Soundness:** 3
**Presentation:** 3
**Contribution:** 2
**Rating:** 6
**Confidence:** 3

**Summary:**

The paper introduces a novel method to assess the difficulty of learning in sequential decision-making problems. This is achieved by conceptualizing the learning problem as the cumulative optimal transport distance between consecutive occupancy measures, which correspond to the policies generated by a learning algorithm. The authors utilize the Wasserstein distance between occupancy measures to define the notion of distance between policies within a manifold. This leads to the definition of the hardness of the learning problem, termed as the Effort of Sequential Learning (ESL), and its connection to regret via the Optimal Movement Ratio (OMR). The authors also present finite-sample estimates for these quantities, with corresponding approximation guarantees. The efficacy of their metrics is demonstrated on classic reinforcement learning (RL) tasks, incorporating a variety of popular RL algorithms.

**Strengths:**

- The approach is innovative and valuable as it effectively incorporates the exploratory components of the learning problem, successfully capturing these aspects.
- The methodology is both theoretically robust and practically applicable. Experiments conducted on grid world and mountain car environments suggest that the proposed methodology can capture key aspects related to exploratory dynamics within the learning problem.
- The paper is clearly written and easy to follow.

**Weaknesses:**

- The main limitation of this work is found in the assumption made in Proposition 1, where the authors presume the existence of the inverse of $P^{\pi}(s,s')$. This assumption effectively implies that any policy $\pi$ is reversible within the environment, meaning the agent following such a policy can always undo its actions. This is a very strong assumption; for example, in an environment where an agent can break a vase, this assumption suggests the agent can also reverse this action and restore the vase to its original state. This assumption is not highlighted enough and significantly narrows the scope of this work in comparison to broader decision-making settings. I recommend the authors explicitly state this assumption and discuss its implications.

- The practical utility of the proposed metrics is unclear. The 3D plots (Fig 3 and 4) are challenging to interpret. I suggest the authors present the same findings using separate 2D plots—one for the distance to optimal and another for the stepwise-distance.

**Questions:**

- The plots in Fig 3 and 4 are tough to parse - what is the big line in Fig 3 for SAC and DQN. Is this the time it takes to fill the replay buffer? If so, why are the colors of the line changing?

---

> ### Author Response · Authors · 2024-11-18
>
> We thank the reviewer for appreciating the problem of interest and our contributions. While we elaborate on the utility of ESL and OMR in the general comment ("Utility of proposed metrics"), we respond to other specific questions here.
>
> **1. Invertibility of Transition Matrix.** First, we would like to clarify that $P^{\pi}(s,s')$ denotes the transition matrix $\mathbf{P}^{\pi}$ rather than an entry in the matrix. We will modify the theorem to clearly state "the transition matrix $\mathbf{P}^{\pi}$". Second, this means that the 'reversibility' between individual states is not needed, and entries in the transition matrix $\mathbf{P}^{\pi}$ can be zero. As long as the transition matrix has an inverse, the occupancy measure space is a differentiable manifold. Otherwise, it is a non-differentiable topological space. Eqs. (24) - (28) shows this concretely. We would also like to highlight that the prior works on linear programming and occupancy measures for MDPs [1,2,3] require existence of similar inverse forms of the transition matrix.
>
> **2. Interpreting Figures 3 and 4.** The plots aim to visualize policy updates and the corresponding dynamics in the occupancy measure space induced by the exploratory processes. More scattered successive points demonstrate large policy updates. The experimental plots Figs. 3 \& 4, showing step-wise distance and distance-to-optimal versus update count (UC), should be interpreted keeping Fig. 2 in mind. Close-by successive points forming a line indicate unchanging or little changing policies - possibly 'stuck in suboptimality'. This can occur even when the policy model is updating e.g. case of ($\epsilon=1$)-greedy. The reviewer is right that for SAC and DQN, the *big lines are due partly to waiting to fill up the replay buffer but also due to 'stuck in suboptimality'*. We will update the figures with the replay buffer collection removed in the revised pdf.
>
> **3. 2D plots for the distance-to-optimal and another for the stepwise-distance.**  2D versions of plots in Figs. 3 \& 4 have been presented in Appendix D.2 for algorithms PSRL, UCRL, DQN, DDPG and SAC, while Appendix D.1 provide enlarged versions of plots in Figs. 3 \& 4. In our revised version, we will also have all 2D plots corresponding to Figs. 3 \& 4 top rows, side by side for comparison, in the appendix.
>
> [1] M. L. Puterman. Markov decision processes: Discrete stochastic dynamic programming. John Wiley and Sons, 1994.
>
> [2] U. Syed, M. Bowling, and R. E. Schapire. Apprenticeship learning using linear programming. In Proceedings of the 25th International Conference on Machine Learning, pp. 1032–1039. PMLR, 2008.
>
> [3] P. N. Ward. Linear programming in reinforcement learning, 2021. URL https://escholarship.mcgill.ca/downloads/xs55mh725. MSc thesis.

---

> ### Author Response · Authors · 2024-11-25
>
> Dear Reviewer  GHMN,
>
> We believe that we have fully addressed your concerns with our responses and updated pdf. We hope that you can update your rating if satisfied and / or ask any further questions within the 1 day discussion period remaining.
>
> Thanks.

---

### Author Response · Authors · 2024-11-18
**Utility of proposed metrics : ESL and OMR are complementary to regret and number of policy updates**

We would like to thank the reviewers for acknowledging the strengths of the contribution as well as for their thoughtful comments and efforts towards improving the manuscript. In the following, we highlight general concerns of reviewers. We then address comments specific to each reviewer by responding to them directly.

**Concerns:**

- *Reviewer GHMN:* The practical utility of the proposed metrics is unclear.
- *Reviewer yMaN:* What information do ESL and OMR provide about the learning process beyond that provided by sample complexity / UC and regret?.


**Response.** In ICML 2018, Exploration in RL workshop (https://sites.google.com/view/erl-2018/open-problems) posed the lack of metrics to quantitatively evaluate different exploration methods as an open problem. The reason is that quantities like regret and number of updates (UC) are outcomes of the exploratory processes, and thus reflect only a partial view of the underlying exploration mechanisms. We propose ESL and OMR to complement regret and number of updates as metrics but not to replace them.

**1. Complementarity of ESL and OMR with respect to UC:**

a. **Case 1.** Let us consider two RL algorithms that reach optimality with the same number of updates, i.e. they have the same UC. How would one be able to distinguish the exploratory processes of these algorithms? ESL and OMR are the summary metrics of the policy trajectory during learning (Figs. 1 \& 2). These can reveal which algorithm's exploratory process is more direct versus meandering, smooth versus noisy, or has large versus small coverage area in the policy space (Figs. 3 \& 4, top rows). Therefore, *ESL and OMR quantify with granularity the characteristics of the exploratory process of an RL algorithm for any given environment.*

b. **Case 2.** Let us consider the case when optimality is not reached but the maximum number of updates is attained by two RL algorithms. How would one be able to evaluate the exploratory processes of these algorithms and systematically uncover which exploratory process demonstrate desired characteristics? Looking into the training trajectories of RL algorithms in an environment and corresponding higher/lower ESLs ($\eta_{sub}$, Sec 4.2), we can make a knowledgeable choice of an RL algorithm exhibiting desired characteristics (e.g. high coverage, smooth exploration). We have shown in Sec 4.2 and results in Appendix B.5 that ranking based on suboptimal ESL is aligned with true ESL, and additionally, the visualisation of the training trajectories (Figs. 3 \& 4) can indicate the characteristics of corresponding RL algorithms even when optimal policy is not reached.

c. **Experimental Evidence.** UCRL2 is known to be provably regret-optimal and is designed to continuously explore. SAC does not have such rigorous theoretical guarantees but is known to be practically efficient. In Table 1, by UC, we observe that SAC is significantly suboptimal than UCRL2. But SAC has lower ESL than UCRL2 as its exploration is smoother. Additionally, OMR for SAC is higher than that of UCRL2. They together indicate that SAC takes smoother but larger number of policy transitions aligned to optimal direction for exploration, while UCRL2 exhibits bigger policy changes and in diverse manner trying to cover the environment faster.

**2. Complementarity of ESL and OMR with respect to Regret.** UCRL2 and PSRL have the same order of regret bound [1]. But PSRL leads to smoother policy transitions that are much more orientated towards optimality (as shown in Fig 3), while UCRL2 leads to less smooth policy transitions that do not taper as it approaches optimality. This information is not evident from regret but from corresponding ESLs and OMRs (Table 1).

**3. Insights for Algorithm Design.** Knowing ESL (or suboptimal ESL) and OMR can assist with developing algorithms that emphasize certain exploratory characteristics. We can develop algorithms with grades of coverage or directness, while also being able to visualize this. Ultimately, depending on the environment, we can choose which characteristics of exploratory process are well suited. In contrast, looking only at the final outcomes of RL algorithms like regret and number of updates does not include these nuances.

We will include a summary of above points in the Discussion in the updated version, and upload it before the end of the Discussion period.

[1] I. Osband, B. V. Roy, and D. Russo. (more) efficient reinforcement learning via posterior sampling. In Advances in Neural Information Processing Systems, pp. 3003—-3011, 2013.

---

### Author Response · Authors · 2024-11-22
**Updated pdf of paper. Looking forward to discussion.**

We hope that our responses address the reviewers' questions. We have also updated the pdf (uploaded in openreview) in line with what we mentioned in our responses.

If our answers address your concerns affirmatively, we hope you would reconsider your assessment. If you have any further questions or comments, we are looking forward to discussing further.

---

### Meta-Review · Area_Chair_KStQ · 2024-12-21

**Metareview:**

This paper introduces the Effort of Sequential Learning (ESL) and Optimal Movement Ratio (OMR) metrics to analyze the learning dynamics of reinforcement learning (RL) algorithms by examining the trajectories of state-action occupancy measures. The paper provides theoretical results and empirical evaluations on standard RL environments like gridworld and MountainCar.

The paper is well-written, and the introduction of ESL and OMR offers a novel perspective on RL training dynamics, moving beyond traditional metrics like sample complexity and regret. However, several concerns were raised by reviewers. The primary issue is the robustness of OMR, which is highly sensitive to small changes during training, making it unreliable in practice. Additionally, ESL and OMR are not well-aligned with learning efficiency or performance metrics, casting doubt on their practical utility. The paper also lacks a clear connection between these metrics and the overall performance of RL algorithms. Further, the finite-sample proposition suggests that the number of samples required to approximate ESL and OMR could be excessively large in real-world tasks.

The paper would benefit from addressing these concerns, particularly by refining the OMR definition for robustness and improving the alignment of ESL and OMR with actual learning efficiency. Additionally, more concrete examples showing the practical applicability of these metrics in RL tasks are needed. Though the paper’s contributions are valuable, the usefulness of ESL and OMR remains unclear without further refinement and demonstration of their relevance to algorithm performance.

**Additional Comments On Reviewer Discussion:**

Based on the authors' rebuttals, while they addressed some points raised by the reviewers, significant concerns remain. Unfortunately, two reviewers did not respond, and while the authors clarified certain issues, the paper still requires substantial revisions. The main concerns revolve around the practical utility of the proposed metrics, their discontinuity, sample complexity, and the relationship between the exploratory process and performance, which were not fully resolved. Therefore, despite the authors' efforts, the paper cannot be accepted in its current form and needs significant revisions.

---

### Decision · Program_Chairs · 2025-01-22

Reject